# Toxic behaviour facilitates echo chammber formation: An agent-based modelling simulation of science attitudes based on Spiral of Silence Theory

**Timothy F. Bainbridge**[ID]\*, **Matthew Ryan, Sinéad Golley, Naomi Kakoschke**[ID], **Emily Brindal**

Health and Biosecurity, CSIRO, Adelaide, South Australia, Australia

\* Tim.Bainbridge@csiro.au

## Abstract

The Internet and social media have facilitated the spread of misinformation and the formation of echo chambers online. These echo chambers may facilitate the adoption of false beliefs and associated costs, but the mechanism of their formation remains a matter of debate. Based on Spiral of Silence Theory, sanctions against opposing views in the form of toxic online behaviour may enable not only the suppression of minority views but also the formation of echo chambers as those with suppressed minority views may attempt to find like-minded individuals who they can safely share their opinions with while avoiding toxic reprisals from those with an opposing view. In the current paper, we introduce the Pro- and Anti-Science Opinions Model (PASOM)—an agent-based model where agents decide between a pro- or anti-science view on a single science-based topic. PASOM uniquely allows agents to choose whether to interact toxically or persuasively. Initial simulations showed that toxic behaviour in the model could push agents into echo chambers and drive agents to adopt strong pro- or anti-science views with most agents in all simulations finishing in an echo chamber. Subsequent simulations demonstrated the importance of toxic behaviour in the outcomes by reducing propensity to behave toxically and sensitivity to toxic behaviour, which resulted in concurrent reductions in echo chamber formation. Finally, simulation outcomes were compared to previously reported social media data and were able to successful reproduce outcomes observed in the empirical data. The various results suggest that toxic behaviour and people's responses to it may be important factors in the formation of echo chambers and differences between social media platforms and topics.

**Data availability statement:** All model output files are available from CSIRO's Data Access Portal (Bainbridge, Tim; Ryan, Matt; Golley, Sinead; Kakoschke, Naomi; & Brindal, Emily (2025): PASOM: Toxic behaviour facilitates echo chamber formation. v1. CSIRO. Data Collection. https://doi.org/10.25919/hj39-0229). Code to run the models is available on GitHub (https://github.com/timbainbridge/PASOM).

**Funding:** The author(s) received no specific funding for this work.

**Competing interests:** The authors have declared that no competing interests exist.

## Introduction

The Internet and social media have allowed people to create and disseminate both true information and misinformation more easily. The spread of misinformation—that is, any information that is untrue, inaccurate, or misleading [1,2]–when believed, can lead people to adopt *false beliefs* [e.g., 3,4]. In some cases, these false beliefs can lead to costly behaviour and the avoidance of beneficial behaviours (e.g., vandalism of 5G towers and the avoidance of vaccination [5]). The potential negative impact on behaviour emphasises the need to explore and further understand how misinformation and false beliefs are formed and spread, especially in an online context.

Online echo chambers might serve as a conduit between misinformation and false beliefs. Echo chambers are networks of interconnected people who share the same or similar views [6,7]. They are much more common online due, in part, to the ease of finding like-minded people or groups [8] and disconnecting from those with conflicting opinions [9,10]. Once formed, echo chamber members may assume that each piece of information shared within the chamber is reliable when, in fact, it is biased by their selection into the chamber [11,12]. With repeated exposure to seemingly confirmatory information, the views of the echo chamber may be adopted and when these views are incorrect, a false belief may be formed [12–14].

### Toxic behaviour and the Spiral of Silence

To date, much of the research on echo chambers has focused on identifying or characterising them based on analysis of social media data [8,14–16]. However, it is arguably as important to understand how they form, as knowing how they form could help with strategies to limit their creation or spread. Past research has examined this question, with aspects of social media platform design (e.g., feed algorithms [17]), network dynamics [18,19], psychological processes (e.g., cognitive biases [20]), and sociological processes [21] implicated.

The *Spiral of Silence* Theory [22] may provide some insights into how echo chambers form. The Spiral of Silence Theory proposes that, for any controversial topic, people assess whether or not their view is in the majority and, if not, they do not express their view due to fear of isolation. In turn, this means that others will be more likely to perceive their view as a minority view, and they too will not express their opinion. This 'spiral of silence' is theorised to result in the suppression and marginalisation of the perceived minority opinion [22,23]. In the online context, the Spiral of Silence Theory would predict similar minority view suppression on public forums [23], but also may push people to find spaces that allow the safe expression of their minority view, namely, echo chambers [24,25].

The original mechanism for the Spiral of Silence's effect was a fear of isolation [22]. However, it has been proposed that the mechanism is *expected sanctions* [23,26], where those in the perceived minority do not speak due to the expectation of sanctions from the those who hold the perceived majority opinion. Although the term 'expected sanctions' is not typically used outside of Spiral of Silence research, in the online context, expected sanctions may take the form of toxic online behaviour. Toxic online behaviour can be defined as, "comments… expressing

disrespect for someone… using insulting language, profanity, or name-calling; by engaging in personal attacks; and/or by employing racist, sexist, and xenophobic terms" [27, p. 924]. Toxic interactions on social media platforms, including hate speech [28,29], have been researched extensively [27,30–32] and the research suggests it can affect the way that people choose to use and interact in online environments [30] with some choosing to avoid engaging on the platform [33] or particular topics [34], while others choose to disconnect from people who act toxically online [35,36]. The ability to disconnect from online communities in response to toxic interactions could mean that sanctions against opposing views, consistent with Spiral of Silence Theory, could be interacting with social media architectures to facilitate the formation of echo chambers online.

The conclusion that toxic behaviour could facilitate echo chamber formation has recently been challenged by Avalle and colleagues [37] who demonstrated that there is no consistent relationship between toxic behaviour and activity on a social media thread. Their results were based on a wide variety of topics and platforms spanning a timeframe from the 1990s to the 2020s. However, as the authors themselves note, the range of correlations between user activity and toxicity (from well below −.5 to well above.5) did not indicate a lack of effect of toxicity on engagement, but instead that a simple story of toxicity reducing activity is far from universal. Nevertheless, a number of limitations in their methods prevent a firm conclusion. First, it is possible that their results were confounded by a third, unmeasured variable. For example, it may be that perceived importance of the topic increased activity, including toxic activity, and the toxic behaviour (or expectations of toxic behaviour) reduced activity but generally not by enough to result in an obvious effect. Second, as their data were analysed by thread, not individual user, it is possible one group of individuals replaced others as the conversations became more toxic, which resulted in small correlations in aggregate. The latter could occur if toxic conversations attract "trolls" who enjoy such conversations, replacing others or unbalanced toxic activity could discourage activity unevenly for opposing opinions, resulting in increased activity on one side and a decline in the other side. Given that Avalle and colleagues [37] did not measure topic interest and did not distinguish toxic behaviour attacking one side from that of the other, neither of these possibilities can be ruled out based on their results. Alternatively, it is possible that the effect of toxic behaviour does not function at the thread or conversation level, instead reducing engagement over time, with little effect within conversations. Given evidence of individual-level effects of greater toxic behaviour on subsequent reductions in user activity in other research [33], the results of Avalle and colleagues [37] are interesting and informative, but do not rule out the potential of Spiral of Silence mechanisms to inform engagement on social media or the formation of echo chambers.

## Modelling opinion and toxic behaviour

One approach that has been used to examine echo chamber formation is modelling. Modelling forces researchers to make their theories explicit with equations that define the model's processes. It also allows theoretical mechanisms to be tested and hypothetical scenarios to be explored by changing a mechanism or parameter within the model [38,39]. Additionally, many social systems involve interactions between individual-level processes, group-level processes, and structural or situational processes, and interactions these systems are difficult or impossible to explore in surveys or social media data [39]. Models can include all of these processes and can be compared to theoretical predictions or observational data [39].

The models used to explore echo chamber formation are often a type of model known as opinion dynamic models [40] within the broader category of agent-based models (ABM) [41–44]. In general, ABMs model agents, representing individuals or organisations, and their interactions. Agents interact with each other and the environment according to rules that specify exactly how these interactions occur. Interactions are repeated over many rounds and changes in agents or other outcomes are noted and explored [41–43,45]. Opinion dynamic models additionally allow agents to share their opinion with their connections, who update their own opinions in response. Opinion sharing is sometimes done directly, with interacting agents' opinions shifting closer to or further from each other, but can also be done indirectly as agents either

observe their connections' behaviour or share information based on their opinions [18–20,46,47]. A subset of opinion dynamic models further allows agents to connect or disconnect from other agents based on their opinions or actions, making the networks dynamic [18,19,46].

Thus far, most opinion dynamic models that have explored echo chamber formation have done so through a combination of methods limiting positive interactions between agents with sufficiently different opinions. These models can be broadly classed as *bounded confidence* models [48,49] or probabilistic variants [19,50]. They specify that agents will only consider views that are within a particular range of their own [18,20,51]; that the effect of opposing agents will be reversed (i.e., making agents' views more extreme) [50,52]; or, in some cases, that agents will disconnect from those with a sufficiently different opinion from their own [18,46]. Especially when the network is dynamic, bounded confidence models allow networks to become fragmented and polarised into echo chambers [18,46].

Although bounded confidence may be a reasonable description of some social media feed algorithms (i.e., the algorithms determining what content to show social media users) [51], they are unrealistic in most other scenarios. Bounded confidence models with dynamic networks either allow for very few interactions between agents with opposing views or they make all or most interactions between agents with opposing views negative. Evidence of toxic interactions between opposing groups [37,53] suggests that interactions between people with opposing views occur frequently and, although connections that can be broadly categorised as negative are not unreasonable [54,55], evidence of people being persuaded by civil interactions [14,56] suggests that not all interactions between those with opposing views are negative. Moreover, to date, negative interactions have been restricted to either dyadic interactions and relationships (i.e., an interaction between only two agents) [54,55,57] or a function of the receivers' opinions relative to sharers' opinions [18,37]. These two cases do not match the typical case of toxic online behaviour, whereby people choose to behave toxically, and that behaviour can be seen by anyone viewing the post (even if it is directed at a specific person), regardless of opinion. Therefore, the toxicity of a comment may be largely a function of the poster's intent, not the receiver's perception or the difference between poster and receiver's opinion. Toxic behaviour, with a theoretical basis in Spiral of Silence Theory, may, therefore, be a more plausible mechanism than those previously employed, as agents can interact with those with opposing views and disconnect from those who act toxically because they acted toxically, not because of interpretation due to opinion differences.

Finally, two prior ABMs have been explicitly created to model Spiral of Silence mechanisms (e.g., [25,58]). Nevertheless, while our work has been inspired by Spiral of Silence, our motivation was to examine the emergence of echo chambers in social media, not Spiral of Silence processes themselves. The latter has led us to include elements not included in these prior models, notably, opinion updating and dynamic networks in response to sanctions, neither of which are part of Spiral of Silence Theory.

## The current study

In the current paper, we present the Pro- and Anti-Science Opinion Model (PASOM)—a new opinion dynamic ABM incorporating insights from the Spiral of Silence Theory. As indicated by the name, the model includes agents who attempt to decide between a pro- or anti-science opinion on a single, controversial, science-based topic. In the model, agents can behave constructively or toxically in favour of either position, with toxic behaviour acting as a sanction against future support of the opposing position and encouraging agents to disconnect from those who act toxically. Agents respond to toxic behaviour against each position by becoming less likely to act in support of that view.

Given Spiral of Silence typically considers bipolar issues (i.e., majority and minority views), we adopted a Continuous Opinion, Discrete Action model [CODA, 59], whereby agents have continuous opinions but can only choose to support either the pro- or anti-science position. In the model, agents can be persuaded by those with markedly different views if either the level of toxicity is low enough that they remain connected or high enough that their view is suppressed. Agents do not know other agents' beliefs, nor do they attempt to infer them, they simply respond to agents' actions. As a result,

in contrast to earlier models, agents will never act toxically in the presence of those with opposing views, if they remain silent.

We selected science controversies as the topic of the model (rather than, e.g., politics) because science topics represent a convenient intersection between echo chamber research, which is often concerned with how echo chambers allow misinformation to spread unchecked [17,60,61], and Spiral of Silence research, which requires topics with clear majority and minority opinions. Given that most people tend to adopt the science-supported view in most cases [62–64], science-based topics provide a useful test case for the intersection between these two research fields.

To explore the model, we first selected a set of baseline parameters to mimic a popular and controversial science topic, where agents' views were initially malleable, and examined the trajectories of mean opinions and echo chamber formations, as well as changes in network structures for selected simulations. Baseline parameters were selected to demonstrate the potential of toxic behaviour, via sanctioning and silencing effects, as a mechanism of echo chamber formation. Second, we explored model outcomes for various values of parameters reflecting agents' sensitivity to toxic behaviour and propensity to behave toxically across three levels of agents' starting confidence. Finally, outcomes from these simulations were compared to patterns of empirical results reported by Cinelli and colleagues [14], following the advice of Valensise, Cinelli, and Quattrociocchi [51]. From these procedures we had three research questions:

1. Can the model reliably form echo chambers?

2. How do model outcomes change as sensitivity to toxic behaviour, propensity to behave toxically, and starting priors change?

3. How well do simulation outcomes match outcomes from empirical social media data?

## Model

### Model outline

In PASOM, all agents represent individuals in an online social network. Agents begin located in a network with initial beliefs on the likelihood of constructive and toxic interactions and an initial opinion, defined by a beta distribution over the probability that the pro-science position is true. The model proceeds though a number of rounds, where each round represents a fixed period of time (e.g., a day). Within each round, originator agents share information supporting either one side of the topic or the other (hereafter pro- or anti-science) and activity propagates throughout the network as agents react to the original shared information, deciding which side of the topic to support (if any) and whether to try and persuade their connections or whether to act toxically towards them. Agents then form new connections or sever existing connections based on whether more or fewer connections would increase their utility from subsequent activity. Finally, agents with no connections connect to a random agent as long as there is at least one agent with a view on the same side of 0.5 as their own or within γ of their own opinion if on the other side of 0.5.

### Agent actions

As noted, the model is a CODA model [59], whereby agents have continuous opinions but can only choose between two opposing positions (i.e., pro-science or anti-science). Agents' behaviour is determined with the following payoff functions, which determine which (if any) position agents choose to support each round:

$$U_{ij1} = b_A \lambda_{ij} \pi^*_{Sij} log \left( K_{ij} \pi^*_{Aij1} + 1 \right) - c_T K_{ij} \pi^*_{Tij1} - c_C + f_{ij} + e_{j1} \tag{1}$$

$$U_{ij0} = b_A \lambda_{ij} \left( 1 - \pi^*_{Sij} \right) log \left( K_{ij} \pi^*_{Aij0} + 1 \right) - c_T K_{ij} \pi^*_{Tij0} - c_C + f_{ij} + e_{j0} \tag{2}$$

Equation 1 represents agent $i$'s payoff expectation for sharing pro-science information in round $j$, and Equation 2 represents agent $i$'s payoff expectation for sharing anti-science information in round $j$. Each function includes agent $i$'s expectation of the probability of toxic interactions ($\pi_{Tij1}^*$, $\pi_{Tij0}^*$) multiplied by their number of connections ($K_{ij}$). This factor reduces their payoffs and, therefore, their likelihoods of posting in support of the position. Additionally, the 'spiral' is exaggerated by the presence of expectations over the probability of positive interactions ($\pi_{Aij1}^*$, $\pi_{Aij0}^*$) multiplied by their number of connections. If agents see very few posts supporting their position, then they will expect less benefit from sharing and share less as a result. These expectations are made with Bayesian reasoning based on past observations (see the model documentation for details, https://doi.org/10.31234/osf.io/jr493 [65]).

Each function also includes agent $i$'s opinion ($\pi_{Sij}^*$, see Opinion updating, below), such that agents who are more pro-science get more benefit from expected positive interactions to pro-science posts, and vice versa for anti-science posts. Other variables and parameters include: interest in the topic ($\lambda_{ij}$, see Opinion updating, below); a fixed cost of posting ($c_C$); a random boost or penalty by round for each agent ($f_{ij}$), representing idiosyncratic costs or benefits to posting each round for each agent; and a random boost or penalty by round that affects all agents equally, with different values for each position ($e_{j1}$, $e_{j0}$), representing random changes in information about the topic that affects all agents identically. Finally, the equations include parameters affecting agents' sensitivity to expected positive interactions ($b_A$) and expected toxic interactions ($c_T$). For further details on each of these variables and parameters, see the model documentation (https://doi.org/10.31234/osf.io/jr493 [65]).

Based on the payoff functions, for agent $i$ in round $j$, if $U_{ij1} > 0$ or $U_{ij0} > 0$, then agent $i$ will share information supporting the position with the higher payoff so long as $|U_{ij1} - U_{ij0}| > u$—that is, so long as agent $i$ is not undecided around which position to support—where $u$ is a parameter indicating how different payoffs have to be for agents to stop being undecided. Once an agent has decided to share, they then decide whether to share constructively, in an attempt to persuade their connections to their view, or toxically, in an attempt to suppress the view opposing their position. This decision is based on the following equation, where $p_{Xij}$ is the probability agent $i$ chooses to share toxically in round $j$ (with the $X$ indicating a variable related to toxic sharing):

$$p_{Xij} = \begin{cases} \min\left(1,\ 2c_X\left(\pi_{Sij}^* - 0.5\right)\left(1 - p_{Sij}\right)\right) & \text{for } \pi_{Sij}^* > 0.5,\ \ I = 1 \\ \min\left(1,\ 2c_X\left(0.5 - \pi_{Sij}^*\right)p_{Sij}\right) & \text{for } \pi_{Sij}^* < 0.5,\ \ I = 0 \\ 0 & \text{otherwise} \end{cases}$$

(3)

The probability is based on the proportion of agent $i$'s connections who supported the pro-science position that round ($p_{Sij}$), agent $i$'s opinion ($\pi_{Sij}^*$), and the $c_X$ parameter, which represents agents' general propensity to act toxically. The presence of $p_{Sij}$ in the equation means that the more opposing information an agent sees, the less likely their opinion will be persuasive, so the more likely they are to act toxically. In contrast, if all information supports their own position, then there is no reason for the agent to act toxically. For $c_X > 1$, without a constraint, it would be possible for $p_{Xij}$ to exceed 1, so the value is kept at a maximum of 1. Finally, agents only share toxically if arguing in favour of the opinion they support (i.e., $\pi_{Sij}^* > 0.5$ for pro-science, $I = 1$; $\pi_{Sij}^* < 0.5$ for anti-science, $I = 0$), given it is possible for agents with opinions close to neutral to support either side.

Finally, utility functions determine agents' expected utility from observing pro-science and anti-science posts given their current set of connections. From these functions, an optimal number of connections for the pro- and anti-science positions are calculated. Derivations of these equations can be found in the model documentation (https://doi.org/10.31234/osf.io/jr493_v2 [65]). The equations for agents' optimal number of connections are:

$$K_{i(j+1)1} = \frac{b_{AG}\pi_{Sij}^*}{c_{TG}\pi_{Tij1}^* + c_{KG}} - \frac{1}{\pi_{Aij1} + c_{BG}}$$

(4)

$$K_{i(j+1)0} = \frac{b_{AG}\pi^*_{Sij}}{c_{TG}\pi^*_{Tij0} + c_{KG}} - \frac{1}{\pi_{Aij0} + c_{BG}}$$

(5)

Agents optimal number of connections for pro- ($K_{i(j+1)1}$) and anti-science ($K_{i(j+1)0}$) information are calculated for the next round (i.e., $j+1$) and are functions of their opinion ($\pi_{Sij}^*$), their interest in the topic ($\lambda_{ij}$, see below), their beliefs about the probabilities of constructive ($\pi_{Aij1}^*$, $\pi_{Aij0}^*$) and toxic ($\pi_{Tij1}^*$, $\pi_{Tij0}^*$) interactions, and parameters relating to the benefit of constructive interactions ($b_{AG}$) and having connections ($c_{BG}$), and the costs of toxic interactions ($c_{TG}$) and maintaining connections ($c_{KG}$).

Agents who shared constructively in support of one or other position compare their optimal number of connections for the relevant equation to their actual number of connections. That is, if an agent shared in favour of the pro-science position, then they compare their optimal number of connections for that position ($K_{i(j+1)1}$) to their actual number of connections. If the difference is less than or equal to −1 (i.e., if they have at least one more connection than desired), and if at least one connection acted toxically against the pro-science position, then the agent will disconnect from a connection who acted toxically against the pro-science position in round j, selected randomly with equal weights from any who did. The same procedure is applied for those who shared in favour of the anti-science position with pro- and anti-science constructive and toxic behaviour swapped. On the other hand, if an agent who shared in favour of the pro-science position has a difference between their actual and optimal number of connections of at least 1, then they connect with an originator agent who was in the same sharing chain who supported that view. It is possible that all originator agents shared the opposing view, in which case the agent would not create any new connections. Again, the same procedure applies in reverse to agents who shared an anti-science opinion.

## Opinion updating

Based on the information they observe, agents attempt to work out which of the two positions—pro-science or anti-science—to believe. As noted earlier, the model is a variant of a CODA Model [59], which means that agents' behaviour is binary (i.e., they can either support the pro- or anti-science position) and they do not have access to their connections' underlying beliefs. As a result, agents' opinions cannot be updated to be closer to or further from connections' opinions. Instead, opinions are updated according to Bayes Rule [66], following Martins' advice for this type of model [59]. Applying Bayes Rule in the current model means agents have a starting belief—a *prior*—represented by a distribution over the probability that the pro-science position is true. Priors are updated to final beliefs—*posteriors*—as agents receive new information from their connections, and the posterior of one round serves as the prior of the next. This approach means that agents will progressively become more difficult to persuade as they acquire more information. In the equations above and the results below, prior and posterior distributions were simplified to a median (i.e., $\pi_{Sij}^*$).

Agents are assumed to be poor at evaluating the truth of the information independently, with a mild pro-science bias, represented by a slightly higher weighting to pro- than anti-science information. A pro-science bias was included because a majority of people, in many or most contexts, hold pro-science opinions [62,63] and do not believe conspiracies [64]. No other biases were included.

The equations describing agents' opinion updating are:

$$\alpha_{Sij} = \alpha_{Si(j-1)} + \lambda_{ij}A_{ij1}\theta$$

(6)

$$\beta_{Sij} = \beta_{Si(j-1)} + \frac{\lambda_{ij}A_{ij0}}{\theta}$$

(7)

The $\alpha_{Sij}$ and $\beta_{Sij}$ variables are parameters for a beta function that represents agent $i$'s opinion in round $j$; $A_{ijl}$ is the number of constructive posts agent $i$ saw in favour of position $l$ in round $j$; and $\theta$ is the bias in the model (pro-science if $\theta > 1$; anti-science if $\theta < 1$; neutral if $\theta = 1$). Higher values of $\alpha_{Sij}$ indicate stronger pro-science opinions and higher values of $\beta_{Sij}$ indicate stronger anti-science opinions. Larger values in general mean opinions are more difficult to change.

The median of the beta distributions describing agents' opinions are calculated with the following equation, representing the approximate median of a beta distribution:

$$\pi_{Sij}^* \approx \frac{\alpha_{Sij} - \frac{1}{3}}{\alpha_{Sij} + \beta_{Sij} - \frac{2}{3}}$$

(8)

Once agents reach a threshold of confidence in their opinion, they begin giving less weight to all information and are less likely to engage, indicating reduced interest in the topic. The feature was including to represent a decline in interest in topics after some initial phase (e.g., interest in COVID-19 vaccine effectiveness [67,68]). The reduction is represented by $\lambda_{ij}$ (in Equations 1–2, and 4–7) which takes the value of 1 when agent $i$ is interested and $v$ otherwise. The equation for interest can be found in the model documentation (https://doi.org/10.31234/osf.io/jr493 [65]).

A list of parameters from Equations 1–8 are included in Table 1. The table includes a number of parameters (i.e., $\mu$, $d_T$, and $d_A$), which are described in the model documentation. The model documentation, which also includes additional equations, descriptions, and full tables of variables, terms, and parameters, is available online (https://doi.org/10.31234/osf.io/jr493 [65]).

## Simulation procedure

A set of baseline parameter values and priors were chosen to enable an initial set of simulations to be performed. Values were selected based on $b_A = b_{AG} = 1$, and priors on toxic and constructive share rates of approximately 8% and 40%, respectively. Opinion priors were selected such that agents' views would start distributed relatively evenly across the possible range (i.e., 0–1) but that they would be malleable within the model simulations and somewhat more malleable for those in the middle of the distribution. Other parameters were selected such that around 10–20% of agents would share in the first round (if they saw another agent's post) prior to the addition of $e_{jl}$ and $f_{ij}$, with these rates approximately doubling after the addition of these random variables. Parameter values were fine-tuned to ensure the graph size did not grow too large and to avoid overly lop-sided results whereby everyone adopted a pro- or anti-science belief in the majority of simulations (at least in the baseline simulations).

To assess echo chamber membership, agents were coded as being part of an anti-science echo chamber if their opinion was less than 0.4 (on a possible range of 0–1) and at least 90% of their connections' opinions were less than 0.5. Pro-science echo chamber membership was coded in an equal and opposite way. That is, agents were coded as being part of a pro-science echo chamber if their opinion was greater than 0.6 and 90% of their connections' opinions were greater than 0.5. This means that if all agents adopt a pro-science stance, they are all categorised as 'in a pro-science echo chamber'. Robustness checks on these definitions are reported in S1 File.

Each simulation of the model started with 500 agents and ran for 200 rounds. With the baseline parameter set, testing revealed that 200 rounds were sufficient for outputs to stabilise, while the 500-agent network size was a trade-off between speed and size. A 1000-agent network was also tested (see the Robustness check section below and S1 File). Two-hundred simulations were run for each parameter set so that variance in outcomes with the same parameter set could be observed.

The starting network was a *cluster network*—that is, a network of clusters of agents with a higher probability of being connected to other agents within the cluster than the probability of being connected to agents outside the cluster. The network was randomly generated with 100 clusters of different sizes with a mean cluster size of 5. Within clusters, agents

**Table 1. A list of selected parameters and terms and their descriptions.**

| Parameter or Term | Description | Baseline Parameter Value |
|---|---|---|
| $I$ | The polarity of the information. 1 = Pro-science information; 0 = Anti-science information. | |
| $j$ | The round. I.e., $j=1$ is the first round. | |
| $i$ | The agent ID. | |
| $U_{ijI}$ | The payoff of agent $i$ for sharing information I in round $j$. | |
| $A_{ijI}$ | The number of constructive interactions agent $i$ had for information type $I$ during round $j$. | |
| $T_{ijI}$ | The number of toxic interactions agent $i$ had for information I during round $j$. | |
| $\pi_{AijI}{}^{*}$ | Agent $i$'s expected probability of a connection responding constructively when sharing information of type $I$ in round $j$. | |
| $\pi_{TijI}{}^{*}$ | Agent $i$'s expected probability of a connection responding toxically when sharing information of type $I$ in round $j$. | |
| $K_{ij}$ | Agent $i$'s number of connections in round $j$. | |
| $K_{ijI}$ | Agent $i$'s desired number of connections for information type $I$ in round $j$. Agents always make this evaluation for the subsequent round (i.e., round $j+1$). | |
| $\lambda_{ij}$ | A multiplier for loss of interest for agent $i$ in round $j$. When interested, equal to 1, otherwise, equal to 0.2 (set by a separate parameter but not changed for any reported simulations). | |
| $\alpha_{Sij}$ | The alpha value of the beta distribution describing agent $i$'s belief that the pro-science position is true in round $j$. | |
| $\beta_{Sij}$ | The beta value of the beta distribution describing agent $i$'s belief that the pro-science position is true in round $j$. | |
| $\pi_{Sij}{}^{*}$ | The approximate median of agent $i$'s probability distribution over science belief at round $j$. That is, agent $i$'s opinion. | |
| $e_{jI}$ | A normally distributed variable with mean 0 and standard deviation of $\sigma_e$, indicating a boost or penalty to information $I$ for round $j$. | |
| $f_{ij}$ | A normally distributed variable with mean 0 and standard deviation of $\sigma_f$, with a separate draw for each agent for each round. The variable represents exogenous variation in agents' motivation to engage with the topic from round to round. | |
| $p_{Xij}$ | The probability that agent $i$ acts toxically in round $j$, given agent $i$ has decided to participate in the round. | |
| $q$ | The probability that each agent has of having the option to start a sharing chain each round. | 0.1 |
| $\sigma_e$ | The standard deviation of $e_{jI}$. | 0.15 |
| $\sigma_f$ | The standard deviation of $f_{ij}$. | 0.4 |
| $b_A$ | A parameter affecting agents' payoffs from constructive interactions. | 1 |
| $b_{AG}$ | A parameter affecting agents' utility from the network due to expected constructive interactions. | 1 |
| $c_T$ | A parameter affecting agents' payoffs from toxic interactions. | 1.25 |
| $c_{TG}$ | A parameter affecting agents' utility from the network due to expected toxic interactions. | 1 |
| $c_C$ | A parameter of the fixed cost of interacting (e.g., time spent, or effort expended). | 1 |
| $c_{BG}$ | A parameter affecting agents' utility from the network per connection. | 0.2 |
| $c_{KG}$ | A parameter affecting agents' utility from maintaining connections (per connection). | 0.05 |
| $c_X$ | A parameter affecting agents' propensity to share toxically. | 1 |
| $u$ | A parameter indicating the difference between $U_{ij1}$ and $U_{ij0}$ required for agent $i$ to post (provided one $U_{ijI}$ is positive). | 0.2 |
| $\mu$ | A parameter affecting how quickly agents lose interest. | 500 |
| $v$ | A parameter indicating how much slower agents update their belief after losing interest. | 0.2 |
| $d_T$ | The decay rate of toxic interactions, with 1 being no decay and 0 indicating completely myopic agents. | 0.9 |
| $d_A$ | The decay rate of constructive interactions, with 1 being no decay and 0 indicating completely myopic agents. | 0.9 |
| $\gamma$ | A parameter representing tolerance for opposing views when connecting to a random agent. | 0.1 |

were connected to all other agents in the cluster (i.e., the probability of connections within cluster was set to 1), and, between clusters, each pair of agents had a 2/500 probability of being connected. In the final generated network, agents had an average of 5.9 connections. The network was created with the sample_pref function from the igraph R package [69] (see the code on github.com for further information; https://github.com/timbainbridge/PASOM).

The starting network was kept constant between simulations, so differences in model outcomes were not due to the initial network setup. Alternative starting networks (including a grid or lattice network) were also tested and produced similar results. See the Robustness checks section and S1 File for further details.

During each round, the model proceeds through a number of steps (or submodels), which are as follows:

1. Each agent has a chance (with probability q) of having the option to start a sharing chain. This represents agents who have some reason to share due to factors exogenous to the model (e.g., they see information about the topic from a source other than the social media network). If these agents choose to share in Step 2, they are *originator agents*.

2. All agents choose whether they would share in support of the pro- or anti-science position or not at all if they were part of a *sharing chain*. Sharing chains are created from originator agents through all other agents who choose to share and are connected, directly or indirectly, to an originator agent. Sharing chains can spread via those supporting opposing positions (consistent with [53]). The decision to share or not is based on evaluations of two payoff functions—one for the pro-science position and one for the anti-science position (see Equations 1–2, above). If either is positive and the difference is greater than u, then they share in favour of the one with the higher payoff.

3. Agents who are part of a sharing chain decide whether to share constructively or toxically. If they share constructively, they share in favour of the position they decided to support in Step 2. If they share toxically, they decide to share in opposition to the position opposed to the one they decided to support in Step 2.

4. Agents use aggregations of shares with them to update their opinions, and their expectations of future constructive and toxic behaviour by their connections. They make separate assessments of likely constructive and toxic behaviour for both the pro- or anti-science positions. Agents also update their interest in the topic (see the Opinion updating section, above).

5. Agents choose whether to disconnect from a connection who shared toxically against the position they supported that round. Inactive agents during the round do not disconnect. The decision to disconnect is based on assessments of whether their utility would increase with fewer agents supporting the relevant position.

6. Agents choose whether to create a new connection. New connections can only occur with originator agents who were in the same sharing chain as the agent (or if an originator themselves) and who shared in support of the same position. As for disconnections, connection decisions are based upon a calculation of their optimal number of connections to maximise their utility.

7. Agents with no connections seek a like-minded connection. They connect to a random agent as long as there is at least one agent with a view on the same side of 0.5 as their own or within γ of their own opinion on the other side of 0.5. The connection is chosen randomly from candidates when at least one option is available.

Simulations were performed by taking the starting conditions and running the function that went through all the steps of the model procedure, then taking the conditions at the end of one round as the starting conditions for the next and so on until the end of the final round.

Code for the simulations was created using R [70] and run on an RStudio Server [71]. The igraph package [69,72] was used for network related functions, and functions from fastmatch [73] and openssl [74] were used to aid the running of the simulations. The ggplot2 [75], cowplot [76], ggnewscale [77], paletteer [78], and reshape [79] packages were used for data manipulation and visualisations. The usethis [80] and gitcreds [81] packages were used to aid the transfer of code to GitHub (https://github.com). The code used to run the simulations and produce the outputs can be found at https://github.com/timbainbridge/PASOM [82]. Outputs from the simulations can be found on CSIRO's Data Access Portal at https://doi.org/10.25919/hj39-0229 [83].

## Empirical comparisons

To see how well model outcomes matched empirical data, we followed advice by Valensise, Cinelli and Quattrociocchi [51] and compared results to those of Cinelli and colleagues [14]. The empirical data included four datasets—Facebook data on pro- and anti-vaccination opinions; data from Twitter (subsequently renamed to X) on political leaning on abortion (we use *Twitter* throughout to match prior publications and its name at the time of the data collection); Reddit data of political leaning on the Politics and News subreddits; and Gab data of political leaning. The scenario of the Facebook data closely matched model assumptions—that is, it was on a binary science topic (anti-vaccines versus pro-vaccine)—and therefore was the most likely of the scenarios to be reproduced by the model. None of the other empirical datasets matched our model design. The topics of the Reddit and Gab datasets tracked political leaning and allowed continuous actions (i.e., one can support centrist politic positions) in contrast to PASOM, which tracked beliefs on a single science topic where only discrete actions were possible (i.e., supporting either a pro- or anti-science position). Although the abortion topic on Twitter was a single largely binary topic (although perhaps not a science topic), it was not coded as such in the empirical data. Instead, people's positions were inferred from the political leaning of news outlets of links they shared. Thus, only the Facebook data closely matched PASOM's assumptions, which would likely result in a greater spread of positions across the spectrum in the empirical data than in the simulated data.

Given we were not able to obtain the Facebook and Twitter data from the original authors and the Reddit data can now only be accessed "for the express limited purposes of community moderation, enforcing Reddit community guidelines, and ensuring community member safety" (i.e., not academic purposes, https://api.pushshift.io/signup), we reproduced figures of agents' opinions to the average of their neighbours' opinions to match those of the social media data and compared pattern of results between our results and those of the empirical data.
Results

## Can the model reliably form echo chambers?

The first set of simulations used a set of baseline parameter values (see Table 1). The baseline model was simulated to determine whether the model could reliably segment agents into separate echo chambers with most or all agents within each echo chamber adopting the dominant view within the chamber. Each simulation produced a matrix of 500 agents by 200 rounds for opinion, pro-science echo chamber membership, and anti-science echo chamber membership. These were aggregated by round such that a single mean opinion score and the percentage of agents in each echo chamber type were generated for each simulation for each round and are displayed in Fig 1. Thus, each line represents the mean score for one of the 200 simulations over each of the 200 rounds and the thick black line represents the "mean of means" of all 200 simulations over each of the 200 rounds. Total echo chamber membership (panel C) was calculated by adding the percentages of agents in each type of echo chamber.

The overwhelming majority of agents in most simulations ended up in an echo chamber (panel C, M = 99.6%, 95% CI [99.5%, 99.6%]), which was split between pro-science echo chambers (panel B, M = 62.7%, 95% CI [61.6%, 63.8%]) and anti-science echo chambers (panel A, M = 36.9%, 95% CI [35.8%, 38.0%]). On average, pro-science echo chambers were more popular due to the pro-science bias in the baseline model.

Differences between the starting network, the most extreme simulations and the average simulation (based on mean opinion) can be observed by examining network structures and density plots of agents' opinions and the average of their neighbours' opinions [14,51] for selected simulations. These are reported in Fig 2. The figure shows the absence of echo chambers in the starting network and the presence of extreme echo chambers in all other cases. Therefore, the set of baseline simulations demonstrated that the model can reliably produce echo chambers.

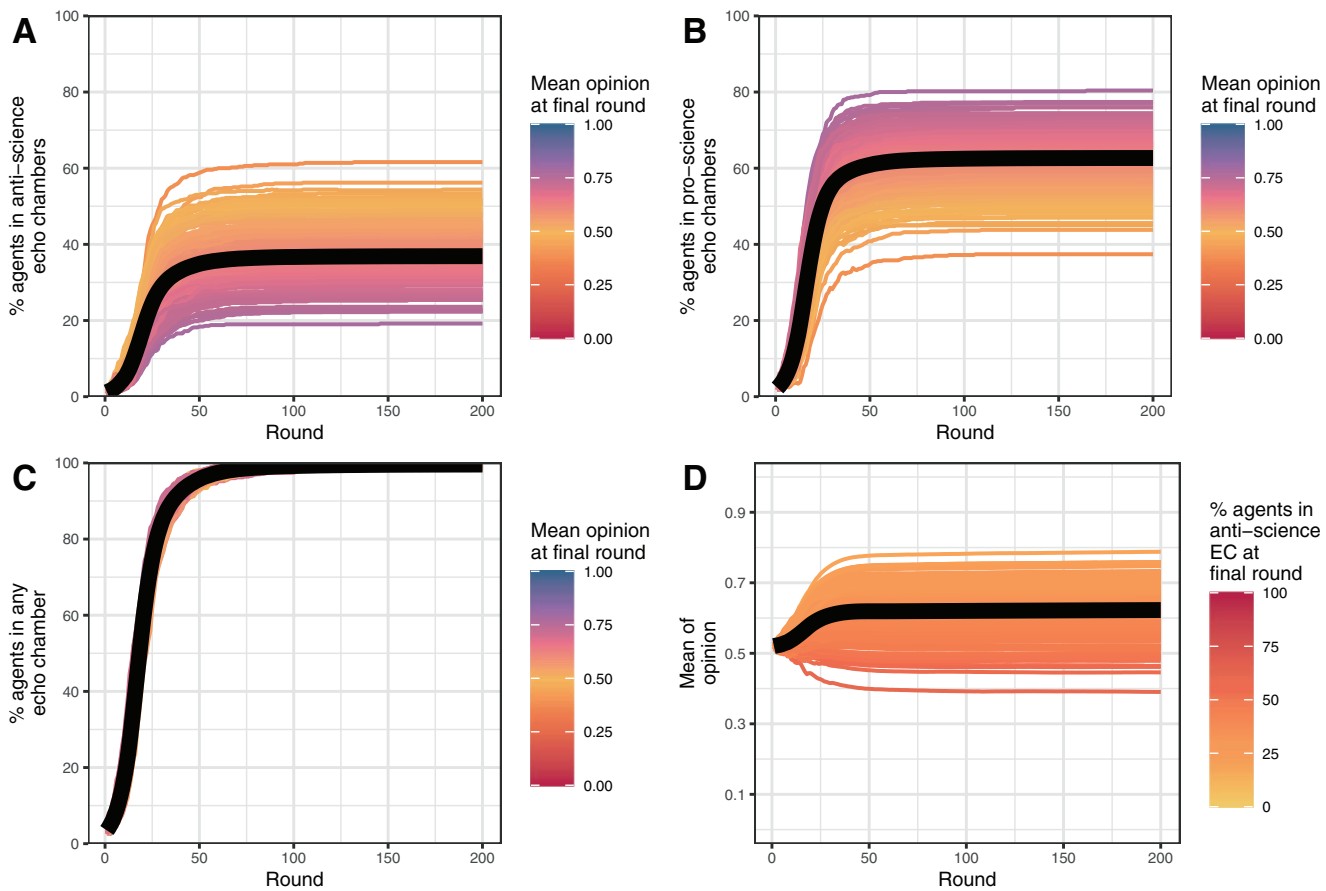

**Fig 1. Mean opinion and echo chamber membership across 200 simulations and 200 rounds with the baseline parameter set.** (A) Percentage in an anti-science echo chamber; (B) Percentage in a pro-science echo chamber; (C) Percentage in a pro- or anti-science echo chamber; (D) Mean opinion. EC = echo chamber. Black line = mean across simulations.

### How do model outcomes change as sensitivity to toxic behaviour, propensity to behave toxically, and starting priors change?

To examine the effect on model outcomes of key parameters and starting conditions, we performed simulations designed to explore the relevant parameter space. That is, given toxic behaviour was a major factor in the creation of echo chambers in the model, we independently reduced both the likelihood of toxic behaviour (i.e., $c_X$ in Equation 3) and sensitivity to toxic behaviour (i.e., $c_T$ and $c_{TG}$ in Equations 1–2 and 4–5, respectively). To simulate overly toxic networks, we even tested $c_T < 0$ to simulated networks where agents prefer toxic behaviour. Finally, initial prior beliefs were also adjusted to prevent opinions from bifurcating as extremely into two groups as in the baseline simulations. Density plots of these simulation are reported in Fig 3 and Fig 4; the percentage of agents who ended up in an echo chamber in each of the simulations are reported in Fig 5 (panels A and B); and mean opinions are reported in Fig 6 (panels A and B).

Lower sensitivity to toxic behaviour resulted in less extreme opinion differences between clusters as agents were more willing to remain connected to those with differing views and these connections moderated the opinions of those in each group. The effect was reflected in lower shares of agents ending in an echo chamber. The effects of changing the parameter were greater at $c_T < 0.5$. At $c_T \geq 0.5$, for a given expectation of the likelihood of toxic reprisal, lower values of $c_T$ made

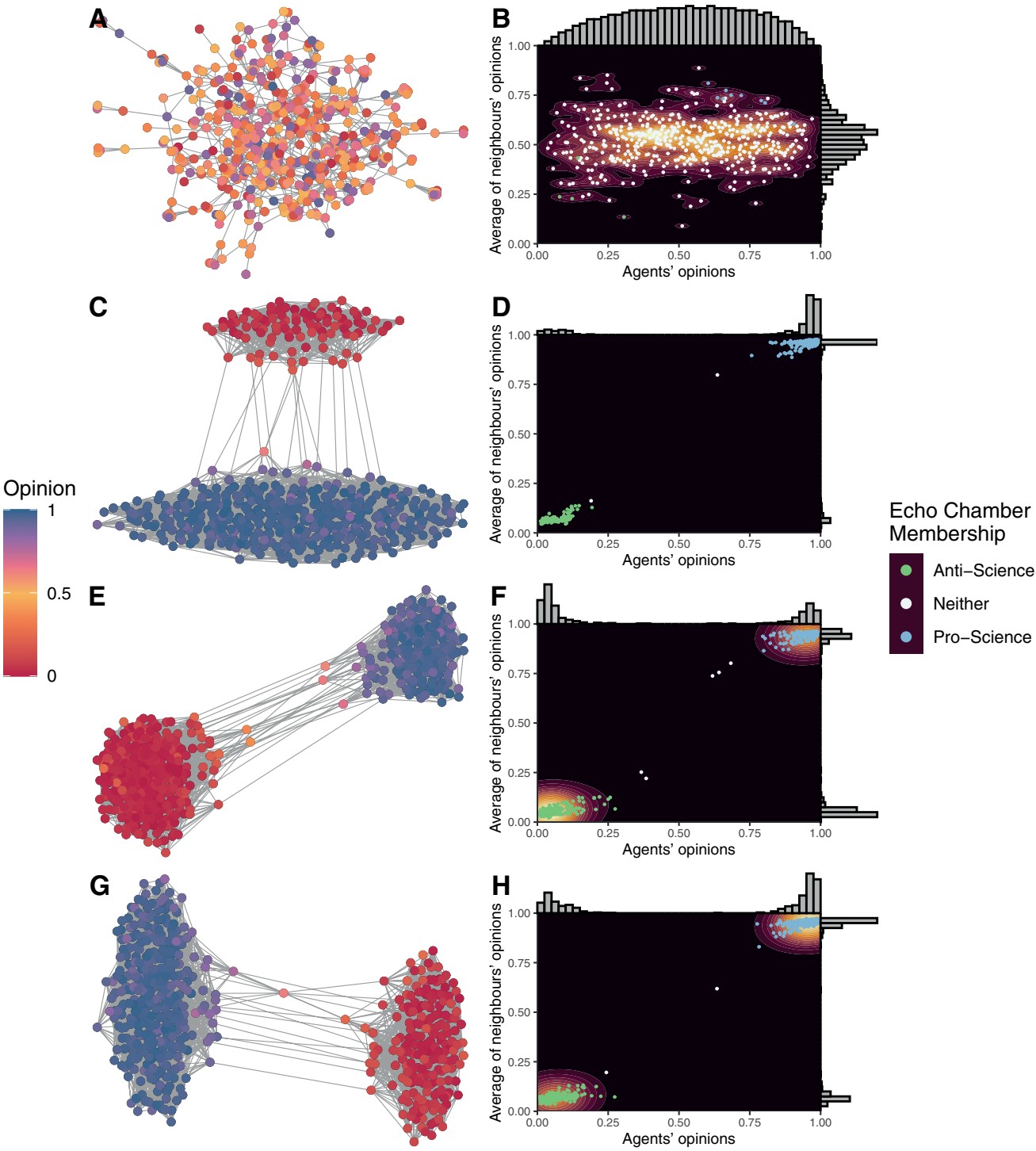

**Fig 2. Example network structures in baseline model simulations.** A = The start of the simulations. C = The end of the most pro-science simulation. E = The end of the most anti-science simulation. G = The end of the simulation closest to the mean opinion of all simulations. Matching density plots of opinions and neighbour opinions in panels (B), (D), (F), and (H). In the network graphs, colour indicates the agent's opinion and lines indicate connections. Layouts were arranged to optimise the network structure. In the density plots, background colour indicates the density of agents with particular opinions and matching average neighbour opinions; points indicate echo chamber membership of the agents.

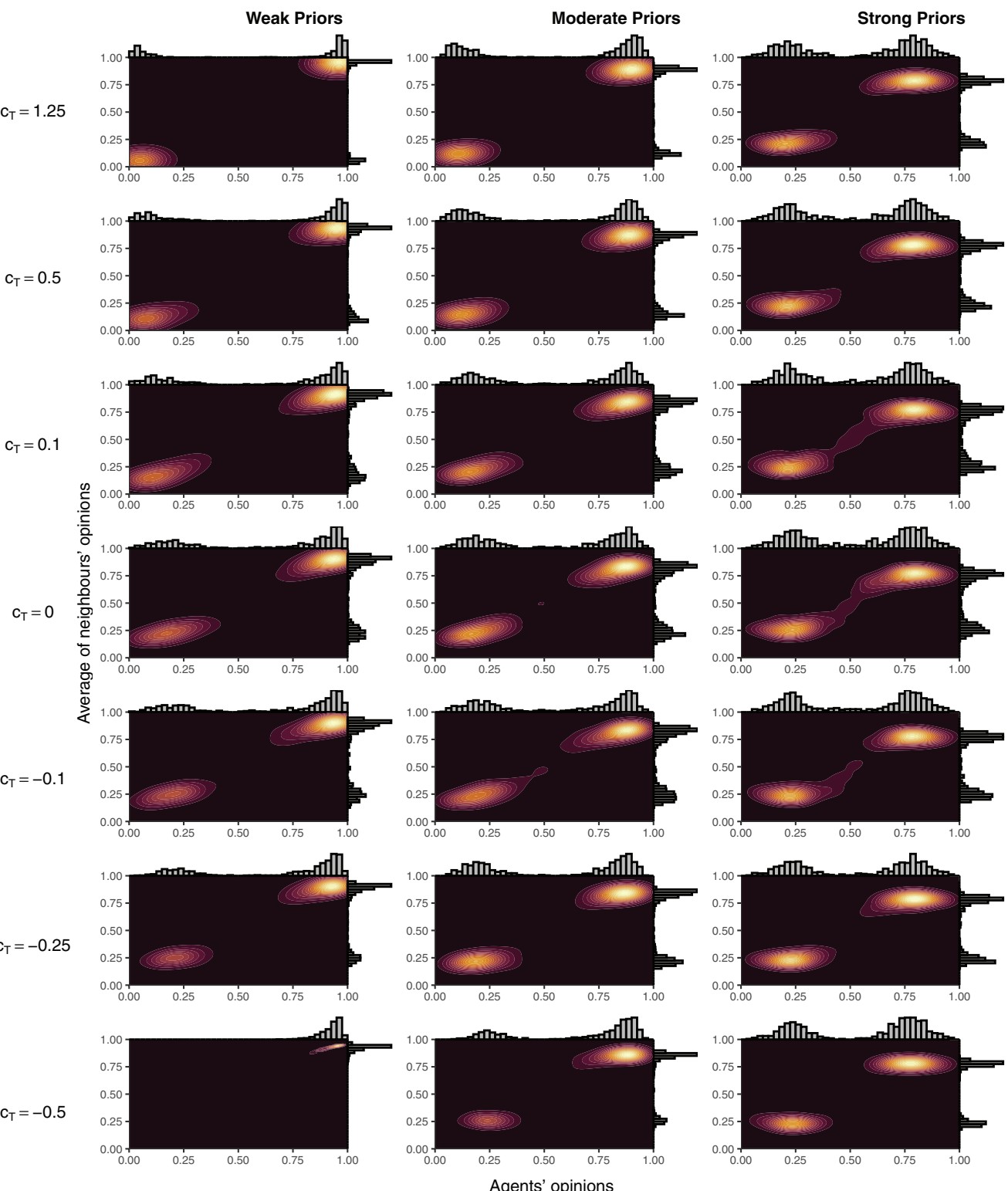

**Fig 3. Density plots of agents' opinions and the average of agents' neighbours' opinions over different values of $c_T$, $c_{TG}$, and the strength of agents' priors.** In all cases, $c_{TG} = 0.8 \times c_T$. For each panel, results are from the 101st simulation (of 200) when sorted in ascending order by the mean of agents' opinions.

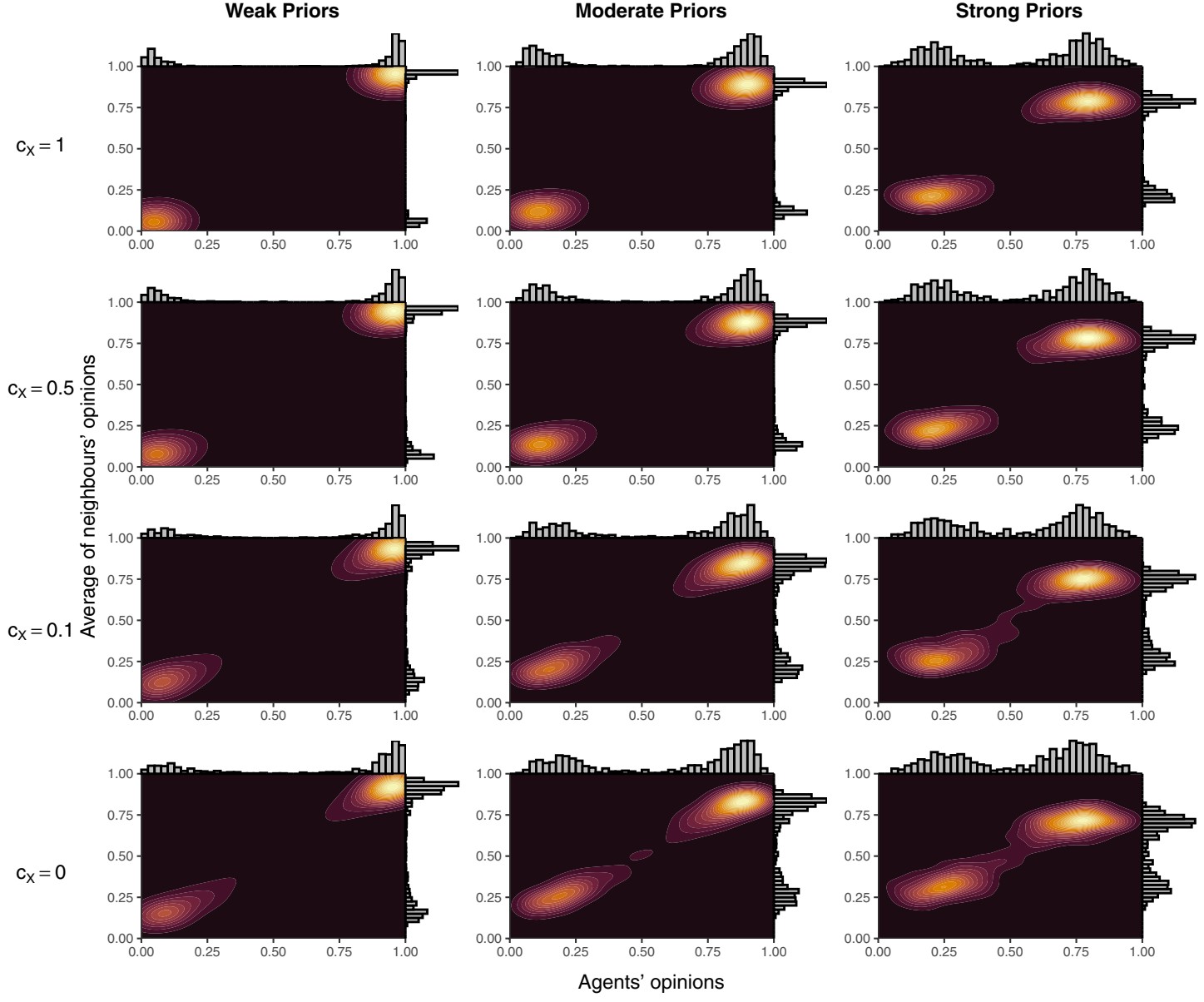

**Fig 4. Density plots of agents' opinions and the average of agents' neighbours' opinions over different values of $c_X$ and the strength of agents' priors.** For each panel, results are from the 101st simulation (of 200) when sorted in ascending order by the mean of agents' opinions.

agents in a local minority more willing to post. In response, the opposing side became more likely to act toxically due to a higher share of agents supporting the local minority view (see Equation [3]). As a result, changes to $c_T$ at higher levels simply adjusted the amount of toxic behaviour while preserving similar behaviour in other respects and, therefore, resulted in similar aggregate outcomes. At $c_T < 0.5$, the balance could not be maintained, and some connections persisted long enough for agents to either be persuaded or remain connected until the final round. At negative values of $c_T$, opposition connections were preferred and, agents became *more* willing to share when connected to those who acted toxically—a reversed Spiral of Silence. Therefore, where disagreement occurred, activity would be heightened, leading to more rapid

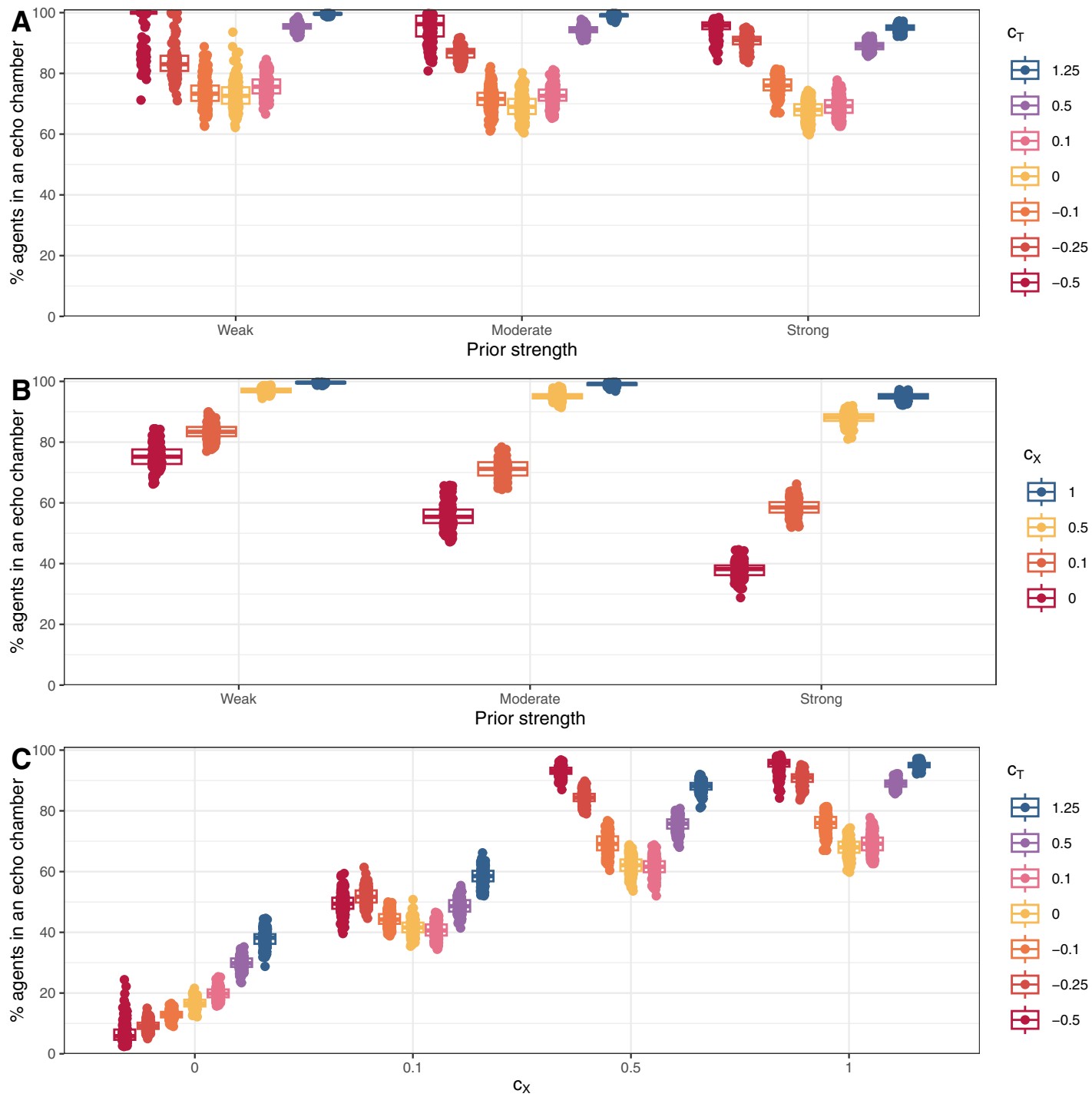

**Fig 5. Boxplots of the percentage of agents in an echo chamber at the end of round 200 for different values of $c_x$, $c_T$, and starting priors.** Models in A used $c_x = 1$; models in B used $c_T = 1.25$; and models in C used strong priors.

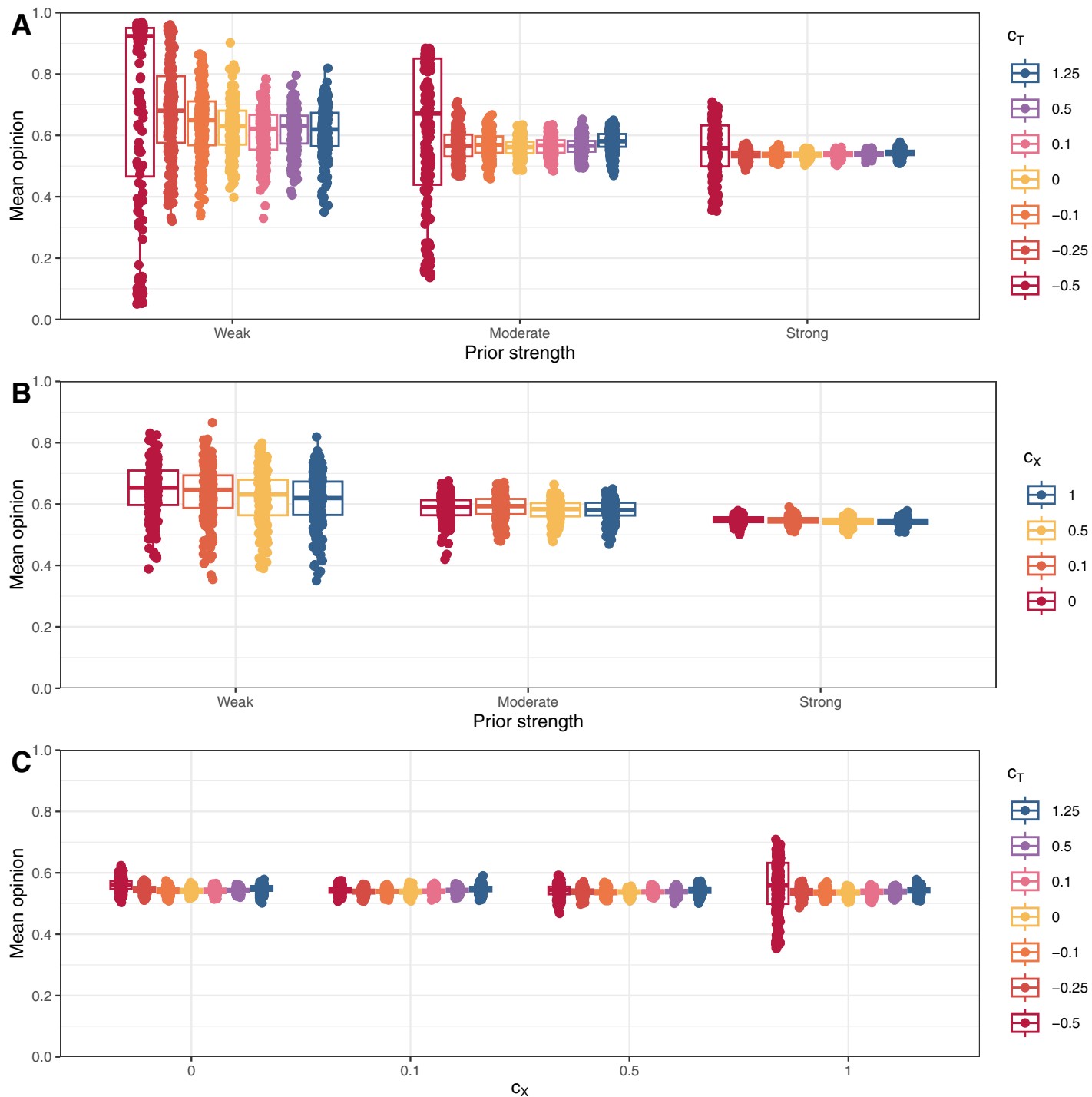

**Fig 6. Boxplots of mean opinions at the end of round 200 for different values of $c_X$, $c_T$, and starting priors.** Models in A used $c_X = 1$; models in B used $c_T = 1.25$; and models in C used strong priors.

persuasion of local minorities. If priors were weak enough, shared connections no longer moderated opinions in different clusters but pulled all agents into a single cluster. These processes were the same but slower for stronger priors and caused the u-shaped pattern in Fig 5A.

Similarly, lower propensities to toxic behaviour resulted in less extreme opinion differences as the incentive and opportunity for agents to disconnect diminished. As for $c_T$, changes were minimal for $c_X \geq 0.5$ as reduced toxic behaviour in any one round would prompt opposition activity, increasing the likelihood of toxic behaviour in subsequent rounds. As a result, the overall effect of changes to the parameter were minimal for $c_X \geq 0.5$. Unlike for sensitivity to toxic behaviour, $c_X < 0$ was not possible (one cannot share less toxically than none at all), so connections to those with opposing opinions could not be encouraged within the model by modifying $c_X$. This meant the complete dominance of one opinion did not occur as for $c_T < 0$. When priors were weak, echo chambers did still form with $c_X = 0$ due to persuasion within local regions and preferential connections with those who shared one's view; however, echo chamber formation never surpassed 85% of agents unlike for the baseline simulations where more than 98% of agents finished in an echo chamber in every single simulation (Fig 1C).

Overall, greater relevance of toxic behaviour—either via greater sensitivity to toxicity or greater propensity to toxic behaviour—resulted in more agents ending up in echo chambers. In these simulations, opposing ideas were either more likely to generate toxic interactions, or toxic interactions had stronger effects on agents' behaviour.

To further explore how these two parameters affected outcomes, we performed a final set of simulations altering both sensitivity to toxic behaviour and propensity to behave toxically. Strong priors were used across all these simulations because we deemed weak priors with negative values of $c_T$ to be unrealistic—those who challenge toxic people on the other side of a debate are unlikely to have weak beliefs. Results from these simulations are reported in Fig 7; the percent of agents who ended up in an echo chamber in each scenario is reported in Fig 5C; and mean opinions are reported in Fig 6C.

Simulations where the effect of toxic behaviour was minimal (i.e., from the centre left and fanning out to the right of Fig 7) resulted in blurred distinctions between clusters. This 'blurring' was reflected in differences in echo chamber membership, with more distinct clusters having more agents in echo chambers. The blurring of results in these simulations occurred when agents would share their opinions freely as either there was minimal toxic activity in response (i.e., when $c_X$ was close to 0) or because they did not care (i.e., when $c_T$ was close to 0). Echo chambers formed in these simulations primarily due to persuasion and the forming of new connections. As both values moved away from 0, Spiral of Silence effects (for $c_T > 0$) or reversed Spiral of Silence effects (for $c_T < 0$) caused the blurring of clusters to diminish. In the case of $c_T > 0$, agents in the middle would either disconnect from those on one side or the other due to toxic activity or they would be silenced against both views, allowing them to be persuaded towards whichever side was more active in their local network. In the case of $c_T < 0$, reversed Spiral of Silence effects caused greater activity in contested parts of the network, meaning agents would tend to be pulled into one of the clusters. In the most extreme of these (i.e., $c_X = 1$ and $c_T = -0.5$), the distribution of mean opinions changed markedly as the extremeness of the reversed Spiral of Silence allowed agents to be pulled into echo chambers of the view they initially did not support, despite strong starting priors. In the simulation reported in Fig 7, this resulted in a relatively even split between pro- and anti-science chambers, but in the most extreme simulations most agents were pulled into one or other cluster. These extreme simulations are reported in Fig 8.

Fig 5C and Fig 7 demonstrated that changes in values of $c_X$ created more substantial differences in outcomes than changes in values of $c_T$ over the parameter space explored with the possible exception of the simulations with $c_X = 1$ and $c_T = -0.5$. Although $c_X = 0$ meant that there was no toxic behaviour, differences in outcomes occurred due to agents' initial expectations of toxic behaviour, which subsided over the opening rounds. These expectations nevertheless caused differences in early behaviour and thus final outcomes as these expectations interacted with different values of $c_T$.

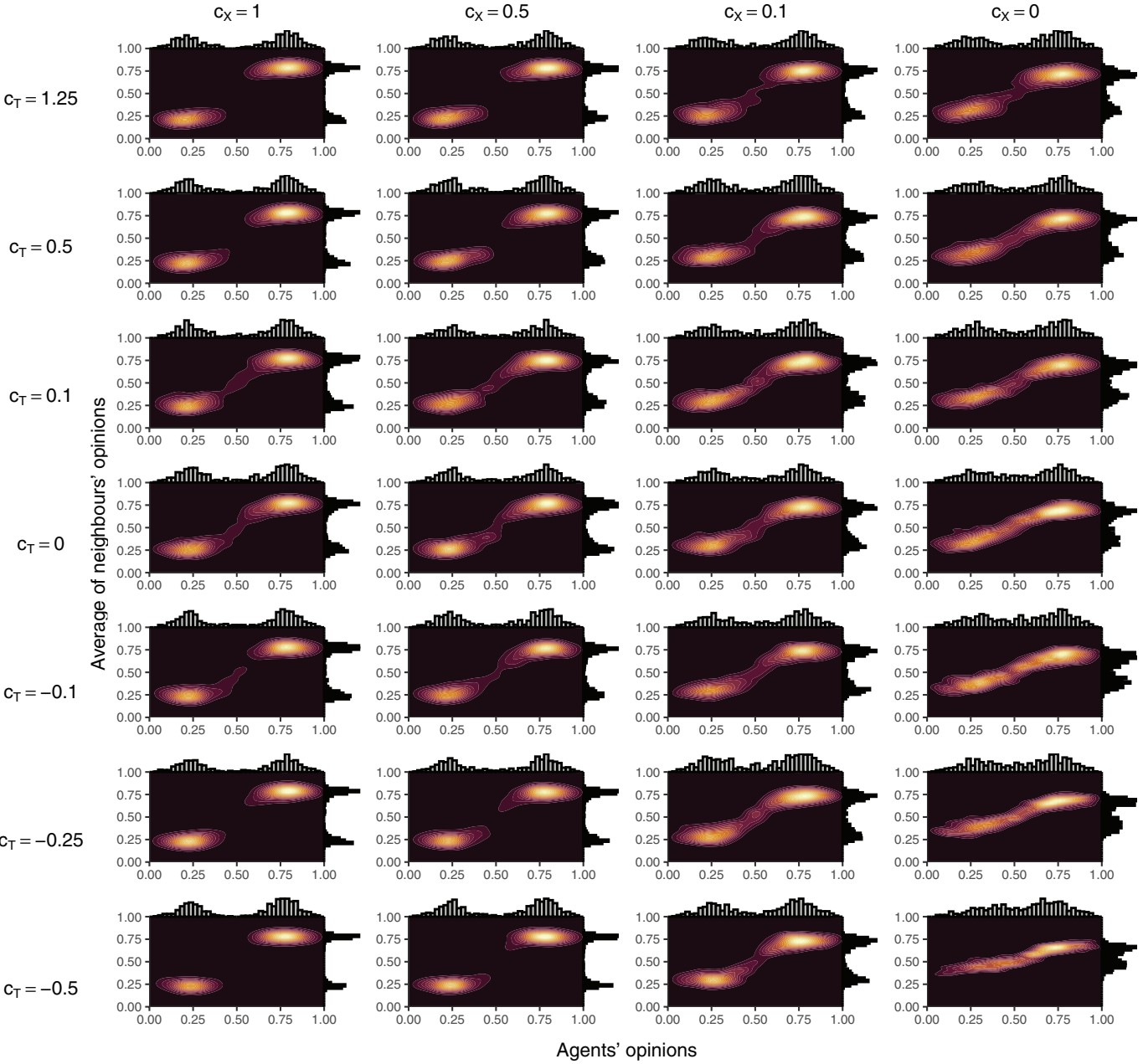

**Fig 7. Density plots of agents' opinions and the average of agents' neighbours' opinions over different values of $c_X$ and $c_T$ with strong priors.** For each panel, results are from the 101st simulation (of 200) when sorted in ascending order by the mean of agents' opinions. The plot in very top left corner matches the values of $c_X$ and $c_T$ from the baseline simulations (although with stronger priors than the baseline simulations).

## How well do simulation outcomes match outcomes from empirical social media data?

To examine model performance, simulation outcomes were compared to equivalent figures produced from real social media data, which have been reproduced in Fig 9 (copied from [14] in line with the CC-BY licence; https://creativecommons.org/licenses/by/4.0/).

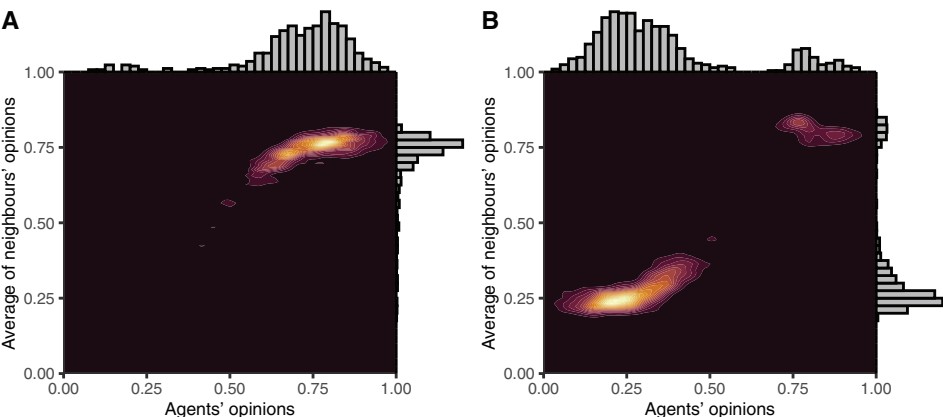

**Fig 8. Density plots of agents' opinions and the average of agents' neighbours' opinions from the most and least pro-science simulations from the set with $c_X = 1$, $c_T = -0.5$, and strong priors.**

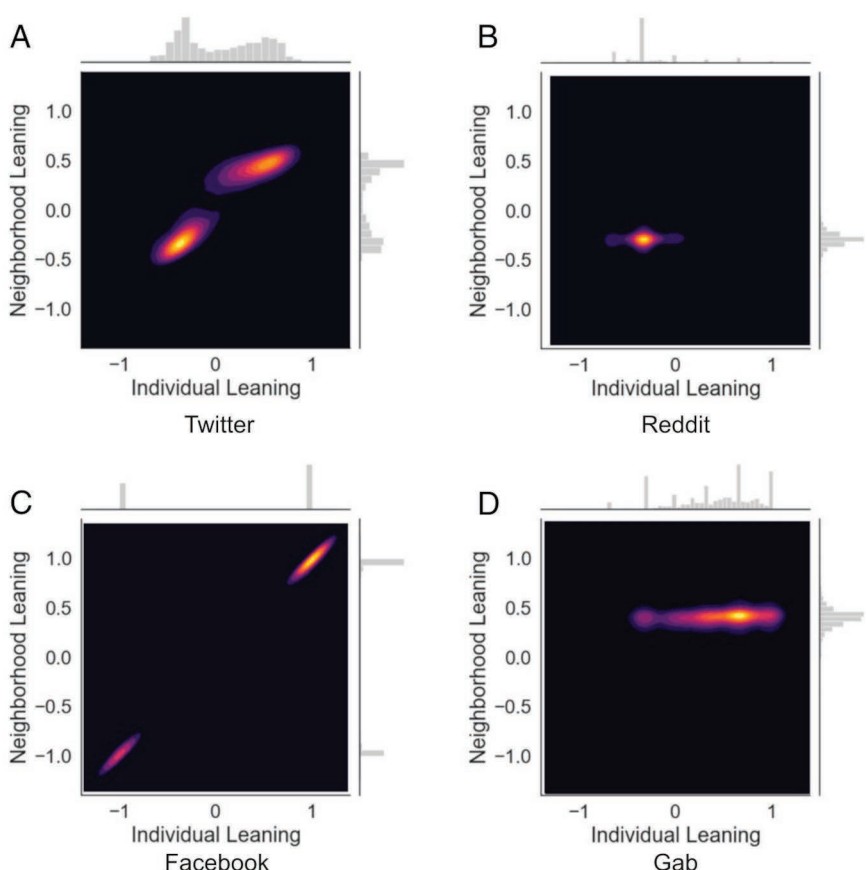

**Fig 9. Density plots of empirical data for various social media datasets.** Copied from Cinelli and colleagues [14], without changes, under the CC-BY licence used by PNAS (https://creativecommons.org/licenses/by/4.0/).

**Facebook.** Facebook results were best replicated by simulations with high sensitivity to toxic behaviour, high propensity to behave toxically, and weak priors (i.e., the top left corner of Fig 3 and Fig 4). In these simulations and the Facebook data, opinions were highly polarised with almost all agents holding extreme opinions and connected to agents with similarly extreme opinions. These matched the assumptions of the baseline simulations and are reflected in the baseline results (Fig 2D, 2F, and 2H). Unlike in the empirical data, the simulations tended to have more pro-science supporters than anti-science supporters (note that anti-vaccine in Fig 9C is positive, i.e., reversed from simulated results), which likely represents a bias in beliefs among those following vaccine-specific pages in the Facebook data relative to the general population, given moderate pro-vaccine support by the public on average [68,84]. However, variations in simulation outputs also allowed anti-science majority simulations to emerge in some cases, matching the empirical data (Fig 2F).

**Twitter.** Empirical Twitter data was characterised by two segregated clusters but, unlike isolated echo chambers, these clusters included moderate agents within each cluster and agents whose average connection was moderate. From Fig 7, the simulations that most closely approximated the Twitter data occurred in models in a band from $c_X = 1$ and $c_T = -0.1$ to $c_X = 0.1$ and $c_T = 1.25$. None of these were perfect, however, as models that tended to have overlap between the X and Y axis in the figures did so via a blurring between the clusters, rather than shifting the clusters together. The combination of parameter sets that fit the Twitter data at least moderately suggest that Twitter users who discuss abortion are either somewhat less prone to toxic behaviour or somewhat less sensitive to toxic behaviour than Facebook users on vaccine pages.

**Gab and Reddit.** Although the Gab and Reddit results look somewhat different (see Fig 9), they each involve a spread of people across a range of opinions, mostly left of centre for Reddit and right of centre for Gab, each connected to agents with views tending to reflect the average of all agents (i.e., consistent with connections across all opinions). Although none of our models from Fig 7 approximated either of these, they were somewhat matched by the simulations reported in Fig 8. Specifically, the most extremely pro-science simulation was a good approximation of the Reddit data, and the most extreme anti-science simulation was a good approximation of the Gab data, albeit with a small cluster of pro-science agents remaining.

**Summary.** Overall, PASOM matched the Facebook data very well; the Twitter data moderately well; and the Reddit and Gab data well but only if the most extreme simulations were selected. Valensise and colleagues [51] compared a model they created and two others (i.e., [50,85]) to the same set of empirical data we have considered here. Their model matched the Facebook data well and another model [50] matched the Twitter data well. However, none of the models matched both the Facebook and Twitter data well and none matched the Gab or Reddit data well. Therefore, PASOM appeared to perform at least as well as the collection of other models examined (although the comparison cannot be quantified without access to the data), despite being designed to match the data generation processes of only the Facebook data.

### Robustness checks

Model outcomes could be heavily influenced by various factors including the starting network or the number of agents. To check that the findings reported were not overly or unexpectedly influenced by these factors, various alternative simulations were employed to check the robustness of the findings. These were not exhaustive and included changes in the starting networks, including a lattice (or 3D grid network), alternative cluster networks, twice as many agents, stronger priors, and various changes to the values of other parameters. While some parameter changes affected outcomes, none of these suggested the results here were not reliable. Most notably mean results did not change substantially for the lattice network nor when the number of agents was doubled. Alternative echo chamber membership definitions were also tested and were found to be either worse at detecting agents in echo chambers or produced similar results. Results of these checks can be found in S1 File.

## Discussion

The internet and social media have changed the way that information is shared and interpreted. These changes have made it easier for individuals to find like-minded connections, facilitating the formation of echo chambers. In some cases, these echo chambers spread misinformation and can cause members to adopt false beliefs, which can have dire consequences for social cohesion or policy interventions on important issues [2–5]. In the current paper, we introduced PASOM—an opinion dynamic ABM of pro- and anti-science opinions based on the Spiral of Silence Theory. PASOM uniquely allows agents to choose between constructive behaviour to persuade others to their view, or toxic behaviour to suppress the opposing view. These actions were facilitated through payoff functions that determined agents' actions.

In the baseline simulations, echo chambers formed readily with almost all agents ending up in an echo chamber. These results suggest that toxic behaviour and Spiral of Silence mechanisms may be important factors in driving people into echo chambers on controversial topics on social media via sanctions and silencing effects.

In subsequent simulations, substantially fewer agents finished in echo chambers when either the propensity to behave toxically or sensitivity to toxic behaviour was reduced. In each case, agents became more likely to preserve connections to agents with an opposing view either because their connection shared toxically less frequently or because they were less sensitive to their connection's toxic behaviour. These connections moderated agents' opinions as they were more influenced by agents with opposing views.

In simulations with $c_X$ or $c_T > 0.5$, outcomes changed only minimally, which may have been due to a 'feedback loop' within the model. That is, as agents became less prone to acting toxically or less sensitive to toxic behaviour, agents became more willing to act, either because they were less sensitive to the toxic behaviour (for $c_T$) or because there was less of it (for $c_X$). In both cases, this prompted greater toxic behaviour in response (due to toxic activity being a function of the amount of activity on the other side, see Equation 3), which, for values greater than 0.5, simply resulted in similar equilibrium outcomes only with greater levels of toxic behaviour for lower values of $c_T$. If this 'feedback loop' is an accurate depiction of online social media usage, then it could mean that interventions to supress toxic interactions online may not effectively translate into lower toxic activity as greater activity in response may simply result in greater toxic behaviour and a new equilibrium.

When sensitivity was set to negative values, agents *preferred* toxic behaviour, which made agents even less likely to disconnect from opposing agents and resulted in a reversed Spiral of Silence where agents became *more* likely to interact when connected to those with an opposing view. If priors were weak, early dominance by one side or the other would routinely allow most or all agents to be persuaded to a single side of the issue. Although these simulations seem unrealistic, results by Avalle and colleagues [37] found positive correlations between toxicity and user activity on some platforms for some topics (e.g., YouTube news comments, Reddit climate change but not conspiracies, Gab, and Facebook news but not vaccines), which could indicate a platform and topic with many "trolls" who may enjoy eliciting toxic responses to their posts [86,87].

Finally, combinations of parameters approximately replicated data from different topics across different social media platforms. In the case of the Facebook data on vaccines, the topic was conceptually well-matched to model assumptions—PASOM assumed the topic was a single-issue science topic, which, broadly speaking, describes vaccines. Moreover, the high value of $c_T$ in the simulations best matched to the Facebook data, appeared to match Avalle and colleagues' [37] finding of a very large negative correlation between user activity and toxicity on Facebook vaccine pages. Although many models have been able to achieve similarly clustered echo chamber outcomes to PASOM, these are almost always achieved through preventing agents from interacting with those with a sufficiently divergent view [18,20,51] or by prompting interacting agents to actively adopt a more extreme opinion [50,52]. Models with these mechanisms are unable to explain a negative relationship with toxic behaviour and user activity and how that may cause echo chambers to form. Therefore, PASOM may represent an incremental step in understanding the formation of echo chambers by offering a potentially more plausible mechanism.

In the case of the empirical data for the other platforms, examinations of the parameter settings were less definitive. The Reddit and Gab empirical results may have been caused by selection effects, rather than negative sensitivity to toxic behaviour, whereby people who disagree with the general views expressed on the platform or page disengage, perhaps due to indirect encouragement by biased content moderation [88]. In the case of Twitter, a wide variety of parameter combinations matched the empirical data moderately well, making it hard to evaluate. Based on Avalle and colleagues' [37] large negative relationships between toxicity and user activity across a range of topics on Twitter (although they did not include abortion as a topic), and Jakob and colleagues' [89] lower rates of toxic behaviour on Twitter than on public pages on Facebook, the models with moderate sensitivity and propensity may be the best match of the Twitter data (i.e., $c_X = 0.5$ and $c_T = 0.5$). If correct, this would suggest that Twitter users posting about abortion are both less likely to post toxic content and less prone to its effects than Facebook users posting about vaccines. However, the plausibility of such a conclusion is difficult to evaluate given the propensity to act toxically does not necessarily translate into greater toxic behaviour if one is insulated from opposing opinions in an echo chamber.

## Limitations and future directions

A major contribution of the current paper is that toxic behaviour can be employed in ABMs to facilitate the formation of echo chambers. First and foremost, empirical validation of the primary mechanisms is required. That is, it is crucial to determine whether and when toxic behaviour facilitates echo chamber formation as people attempt to avoid toxic interactions and whether and when toxic interactions silence people online. Although the latter is plausible, it has not yet been demonstrated that toxic behaviour facilitates echo chamber formation. Specifically, researchers have primarily investigated the actions of individuals in echo chambers rather than examining how interactions with others influence people's progression into echo chambers (e.g., [31]). Moreover, if the silencing effect is a key element, then that would also have to be established by linking reduced activity to prior toxic interactions. Although Avalle and colleagues' [37] research is a useful guide to this process, it does not examine activity at the individual level over time, and hence, it is incapable of providing a definitive answer. Additionally, establishing such a mechanism would further require ruling out the alternative explanation that users progress into echo chambers primarily because of feed algorithms, not to avoid toxic behaviour. Various models have demonstrated the plausibility of this latter explanation [17,51]. In reality, both mechanisms may play a role and models including both in the future could prove fruitful in teasing apart their various effects.

As toxic interactions are a more realistic mechanism to explain human behaviour than bounded confidence, it would be useful to apply the insights from the current paper to a wide variety of opinion dynamic ABMs and closely related models. However, as PASOM is a continuous opinion discrete actions model (CODA), it is not clear how the toxic behaviour of this model could be transferred into more conventional opinion dynamic models where agents have continuous actions, such as sharing their exact opinion with their connections. Incorporating toxic behaviour into other model varieties would require some consideration of how toxic behaviour would affect agents' actions (given the payoff equations, Equations 1–2, only work with discrete actions). Importantly, if this problem can be addressed, it may allow consideration of majority and minority views within particular sub-regions of a larger debate, given evidence of within group toxic behaviour, which may serve to suppress local minority views (e.g., "anarchism vs. state socialism" within left-leaning groups on Reddit [31], p. 205).

In a similar vein, factors that are peripheral to the Spiral of Silence Theory, and thus, not well defined, were required for the current modelling. For example, it was not clear what the function of toxic behaviour should be with respect to opposition posts. Spiral of Silence theory does not specify whether minority view members who decide to share their opinion would be more likely to be conciliatory in an attempt to limit toxic reprisals or more aggressive to try and discourage opposition supporters from retaliating, and other research has tended to focus on minority influence or persuasion (not toxic behaviour) [90,91], or offline activity [92], which may not follow the same processes as online activity [93]. Changing the way this or other parts of the model worked may have changed the pattern of outcomes reported but was beyond the scope of the current work.

In PASOM, agents were largely homogeneous with moderate differences in their starting opinions being their one major difference (see Fig 2B). This meant that, with identical opinions and expectations of likely interactions, agents would process information identically and have an equal probability of sharing information or acting toxically. This contradicts empirical research, which suggests that people process information differently [94–96], share on social media at different rates [97], are affected differently by toxic interactions [98], and interact toxically online more or less frequently [99,100]. Future research could either allow one or more of these parameters to vary or could attempt to incorporate research from personality psychology on the correlates of online social media activity (e.g., [101,102]). Although allowing parameters to vary by agent would make the model more complex, a number of reasons to do so are apparent. First, allowing variability in parameters may lead to more realistic patterns in model outcomes for otherwise similar model setups, which could include qualitatively different patterns of results. For example, if some agents preferred toxic interactions while others did not, simulations may resolve into two extreme clusters of toxic-averse agents, while another set of agents may be spread between the extremes and connected to agents with a wide range of opinions. This may ultimately prove important to understanding real-world social media findings, such as how correlations between toxic behaviour and user activity can sometimes be positive without requiring all agents to at least be indifferent to toxic activity [37]. Second, interesting aspects of model outcomes might be investigated. For example, one might ask which agents are more likely to be 'influencers', or which agents are more willing to stay connected to those with opposing views. Not only could this inform our understanding of social network dynamics, but it could lead to interesting hypotheses about the relationship between personality and social media use.

## Conclusion

In the current paper, we aimed to assess the plausibility of toxic behaviour as a mechanism of echo chamber formation. To achieve this aim, we introduced PASOM—an ABM of pro- and anti-science opinions. Mimicking expected sanctions within the Spiral of Silence Theory [22,23], toxic behaviour suppressed opposition views, but also encouraged agents to sever connections with opposing toxic agents. These features enabled toxic behaviour to facilitate echo chamber formation in the model, suggesting that toxic behaviour could be an important part of echo chamber formation in an online setting. When agents' propensity to behave toxically was reduced, fewer agents ended up in echo chambers; however, echo chambers still formed due to persuasion. When agents' sensitivity to toxic behaviour was reduced, fewer agents ended up in echo chambers initially, but the pattern reversed at low or negative sensitivity values. Specifically, reversed Spirals of Silence caused increased activity in regions of disagreement. The latter effect often resulted in the persuasion of most or all agents to adopt the majority view, although when initial priors were strong it only occurred sometimes. Finally, the simulations successfully reproduced outcomes similar to various analyses of social media data; however, only the Facebook dataset, which was well matched to PASOM's assumption, provided unequivocal results. Overall, the current research suggests that toxic behaviour could be an important factor in explaining echo chamber formation on social media and an effective mechanism for exploring outcomes in opinion dynamics of controversial topics, including those based on science. Further work is required to implement toxic behaviour into other model varieties or to make PASOM mechanisms applicable for a wider variety of real-world predictions.

## Supporting information

**S1 File. Supporting Information.** Robustness checks for changes in starting networks and parameter values not reported in the paper, and the effects of changes to the definition of an echo chamber.
(PDF)

## Author contributions

**Conceptualization:** Timothy F. Bainbridge.

**Data curation:** Timothy F. Bainbridge.

**Formal analysis:** Timothy F. Bainbridge.

**Funding acquisition:** Sinéad Golley, Emily Brindal.

**Investigation:** Timothy F. Bainbridge.

**Methodology:** Timothy F. Bainbridge, Matthew Ryan.

**Project administration:** Timothy F. Bainbridge, Sinéad Golley.

**Supervision:** Sinéad Golley, Naomi Kakoschke, Emily Brindal.

**Validation:** Timothy F. Bainbridge, Matthew Ryan.

**Visualization:** Timothy F. Bainbridge.

**Writing – original draft:** Timothy F. Bainbridge.

**Writing – review & editing:** Timothy F. Bainbridge, Matthew Ryan, Sinéad Golley, Naomi Kakoschke, Emily Brindal.

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
