## [Decision Letter · Decision Letter 0]

15 Oct 2024

PONE-D-24-33055Toxic behaviour facilitates echo chamber formation: An agent-based modelling simulation of science attitudes based on Spiral of Silence TheoryPLOS ONE

Dear Dr. Bainbridge,

Thank you for submitting your manuscript to PLOS ONE. After careful consideration, we feel that it has merit but does not fully meet PLOS ONE’s publication criteria as it currently stands. Therefore, we invite you to submit a revised version of the manuscript that addresses the points raised during the review process.

**The reviewers acknowledge the paper's potential but identify several key areas for improvement. They emphasize the need for clearer explanations of the methodology and stronger comparisons with existing models to establish the paper’s novelty. Also, they recommend validating the model with real-world data rather than relying solely on synthetic simulations and suggest incorporating more references on toxic behavior. Minor revisions regarding text clarity, figures, and parameter descriptions are also required. I strongly recommend to address all these points in your revised submission.**

We look forward to receiving your revised manuscript.

Kind regards,

Fabiana Zollo, Ph.D.

Academic Editor

PLOS ONE

**Journal Requirements:**

2. Please note that your Data Availability Statement is currently missing the repository name. If your manuscript is accepted for publication, you will be asked to provide these details on a very short timeline. We therefore suggest that you provide this information now, though we will not hold up the peer review process if you are unable.

**Additional Editor Comments:**

The reviewers acknowledge the paper's potential but identify several key areas for improvement. They emphasize the need for clearer explanations of the methodology and stronger comparisons with existing models to establish the paper’s novelty. Also, they recommend validating the model with real-world data rather than relying solely on synthetic simulations and suggest incorporating more references on toxic behavior. Minor revisions regarding text clarity, figures, and parameter descriptions are also required. I strongly encourage the authors to address all these points in their revised submission.

Reviewers' comments:

Reviewer's Responses to Questions

**Comments to the Author**

1. Is the manuscript technically sound, and do the data support the conclusions?

Reviewer #1: Partly

Reviewer #2: Yes

2. Has the statistical analysis been performed appropriately and rigorously? 

Reviewer #1: Yes

Reviewer #2: Yes

3. Have the authors made all data underlying the findings in their manuscript fully available?

Reviewer #1: Yes

Reviewer #2: No

4. Is the manuscript presented in an intelligible fashion and written in standard English?

Reviewer #1: Yes

Reviewer #2: Yes

5. Review Comments to the Author

**Reviewer #1: ** The paper introduces a new agent-based model incorporating several features, particularly the propensity for toxic interactions. The authors use this model to simulate social media interactions and explore the emergence of echo chambers and opinion polarization, linking their analysis to the "Spiral of Silence" theory. While the paper presents a technically sound analysis, I have several concerns regarding its relevance and framing.

First, the authors attempt to relate their model to the "Spiral of Silence" theory in the context of Pro and Anti-science opinions. However, the causal relationship between their model and the theory is unclear. Recent research (Avalle, M., Di Marco, N., Etta, G. et al. Persistent interaction patterns across social media platforms and over time. Nature 628, 582–589 (2024). https://doi.org/10.1038/s41586-) suggests that toxicity does not necessarily discourage participation in conversations. As a result, the silencing mechanism proposed by the "Spiral of Silence" theory may not be a suitable explanation for the behavior modeled. Additionally, the inclusion of toxic interaction in the users' payoff function raises concerns, as it is not clear whether this might induce users to express opinions contrary to their beliefs. If this is the case, I believe the model could deviate significantly from reality, making its linkage to social psychology theories inappropriate.

Secondly, the authors state that "to our knowledge, no model has incorporated insights from Spiral of Silence Theory and the concept of toxic interactions to explore opinion progression and echo chamber formation." This claim is only partially accurate. The concept of "hostile interaction” or “negative link” has already been explored in previous models (see, for example, Leskovec, Jure, Daniel Huttenlocher, and Jon Kleinberg. 'Predicting positive and negative links in online social networks.' Proceedings of the 19th international conference on World Wide Web. 2010).

The results of the model also require further clarification, as some outcomes seem implausible when compared to real-world scenarios. For instance, the authors claim that "61.0% of simulations resulted in all agents adopting the same view," which does not align with typical social media dynamics. This raises doubts about the model's relevance to the Pro/Anti-science debate, whether it accurately reflects any real life behavior and thus how it can be used to validate the hypothesized role of “Spiral of Silence” in shaping online debates.

Moreover, the authors chose certain parameters as a baseline without providing sufficient justification. It remains unclear whether the model can reproduce real-life outcomes for specific parameter choices. The authors could strengthen their validation by comparing their results to empirical cases, as seen in Valensise, Carlo M., Matteo Cinelli, and Walter Quattrociocchi. 'The drivers of online polarization: Fitting models to data.' Information Sciences 642 (2023): 119152.

Another critical point requiring more attention is the exploration of the parameter space. As the authors note, they vary the toxicity parameter in increments of 0.5 within the interval [0,2]. However, in the range [1,2], the model exhibits minimal variation, with only one data point in the lower half of the range. I recommend increasing the granularity of this parameter sweep and offering a more in-depth explanation for the model's behavior. Moreover, the statement that "this may have been because toxic propensity was sufficient for agents' views to effectively suppress opposing opinions for any value of cX ≥ 1" warrants further exploration. This analysis could provide insights into the model's transition behavior and the underlying cause of such changes.

Lastly, Figure 1 has an incorrect ordering of the plots (A, C, B, D), making it difficult to follow. I suggest correcting this to improve clarity.

**Reviewer #2: ** This paper presents a new model for echo chamber formation, grounded in the Spiral of Silence Theory. According to this theory, when faced with a controversial topic, individuals assess whether their opinion aligns with the majority. If they perceive their view as being in the minority, they refrain from expressing it due to a fear of social isolation.

The authors implement a very detailed mathematical model upon these premises, which according to the simulations, is capable of recreating echo chambers.

Although I found a personal interest in how the model has been constructed and implemented, I think that, at this stage, the work needs to be improved in many aspects, which I summarize below.

- In some parts, the paper is not very clear about the steps followed by the authors to obtain the results;

- Even if the model is interesting, it is very complex and contains a lot of parameters to set. Since there are a lot of works proposing models of echo chamber formations, I’m not sure that this paper adds some novelty. To convince the readers that your model is better than others, you should add some comparisons with other models, if possible. For example, you could consider these works (and add them to the bibliography if needed):

https://www.sciencedirect.com/science/article/pii/S0020025523007375

https://journals.aps.org/prl/abstract/10.1103/PhysRevLett.124.048301

https://www.sciencedirect.com/science/article/pii/S0020025521012901

- Similarly to the previous point, this work has a significant limitation: the model is tested solely on synthetic data, with no validation on real-world data. While simulations can offer valuable insights, testing on real data is essential to demonstrate the model's practical effectiveness. If possible, you should carefully consider a methodology similar to the one presented in https://www.sciencedirect.com/science/article/pii/S0020025523007375

- The paper lacks some important bibliographic information about toxic behaviour, for example:

https://www.sciencedirect.com/science/article/pii/S1359178921000628

https://journals.plos.org/plosone/article?id=10.1371/journal.pone.0278511

https://www.nature.com/articles/s41586-024-07229-y

Some minor remarks:

- In the Introduction, sometimes the text is hard to read due to a high number of repetitions. One example is in lines 29-32, where ‘echo chamber’ is repeated several times. Please improve the text where necessary.

In line 168, the authors explain when an agent decides to share information supporting one of the two sides. Since the difference between the utilities could be (possibly?) small, have you considered using the difference between the two payoffs instead of their value? In this case, an agent decides to share information only if the difference between the two payoffs is greater than a certain threshold. This could avoid some cases in which the two payoffs are too similar to make a real decision.

- At line 173, in p_{Xij}, what X stand for? Moreover, have you considered setting different C_x for each user? While I recognize that this would increase the complexity of an already intricate model, it could potentially lead to more nuanced simulations.

- Line 197-202: please try to explain better this part. The mechanism underlying network updates is not clear, at least to me.

- Have you tried to define Echo Chambers using different thresholds? Is your procedure robust? (Linea 245-249)

- Please add more details on how the starting network was generated.

- Line 286: maybe it’s better to write 200 rounds instead of simulations.

- Why are the labels in Figure 1 not alphabetically ordered?

- Please delete the C_x color legend from Figures 3, 4 and 6. The color is already clear from the x-axis.

- Line 350: maybe you mean “presented in Fig. 5”?

6. PLOS authors have the option to publish the peer review history of their article (what does this mean? ). If published, this will include your full peer review and any attached files.

**Do you want your identity to be public for this peer review?** For information about this choice, including consent withdrawal, please see our Privacy Policy .

Reviewer #1: No

Reviewer #2: No

---

## [Author Response · Author response to Decision Letter 1]

5 Mar 2025

To the Editor and Reviewers,

We are very grateful for the reviewers’ comments and believe the paper is much better as a result. Thank you for your time and consideration.

We have reproduced reviewers’ comments below with our responses following each comment.

The reviewers acknowledge the paper's potential but identify several key areas for improvement. They emphasize the need for clearer explanations of the methodology and stronger comparisons with existing models to establish the paper’s novelty. Also, they recommend validating the model with real-world data rather than relying solely on synthetic simulations and suggest incorporating more references on toxic behavior. Minor revisions regarding text clarity, figures, and parameter descriptions are also required. I strongly encourage the authors to address all these points in their revised submission.

Response:

Reviewers’ comments on uncertainty around the realism of the results and the suggestion to map outcomes to empirical data have led us to make minor changes to our model, moderate changes to starting conditions and parameter values, and to add a comparison of our simulations to prior empirical results derived from social media data. These changes have necessitated extensive changes throughout the manuscript.

Reviewers' comments:

3. Have the authors made all data underlying the findings in their manuscript fully available?

Reviewer #1: Yes

Reviewer #2: No

Response:

All the data are synthetic and were generated by code. Given we had made the code available, we initially chose not to post the data itself. However, we have decided to post the output data from the simulations, which are available at: https://doi.org/10.25919/hj39-0229.

5. Review Comments to the Author

Reviewer #1: The paper introduces a new agent-based model incorporating several features, particularly the propensity for toxic interactions. The authors use this model to simulate social media interactions and explore the emergence of echo chambers and opinion polarization, linking their analysis to the "Spiral of Silence" theory. While the paper presents a technically sound analysis, I have several concerns regarding its relevance and framing.

First, the authors attempt to relate their model to the "Spiral of Silence" theory in the context of Pro and Anti-science opinions. However, the causal relationship between their model and the theory is unclear. Recent research (Avalle, M., Di Marco, N., Etta, G. et al. Persistent interaction patterns across social media platforms and over time. Nature 628, 582–589 (2024). https://doi.org/10.1038/s41586-) suggests that toxicity does not necessarily discourage participation in conversations. As a result, the silencing mechanism proposed by the "Spiral of Silence" theory may not be a suitable explanation for the behavior modeled.

Response:

Thank you for notifying us of this paper. We agree that it calls into question the relationship between toxicity and engagement on social media platforms and challenges the general view that all toxic behaviour always causes lower activity. However, we do not believe the evidence in the paper allows the rather stronger point that toxic behaviour does not inhibit activity at all and therefore believe the silencing effect could be important in some instances. Consequently, we have added a paragraph on the above paper to the introduction and added simulations whereby we alter sensitivity to toxic behaviour.

For convenience, we have reproduced that paragraph here, which can be found on page 5, line 68:

“The conclusion that toxic behaviour could facilitate echo chamber formation has recently been challenged by Avalle and colleagues [37] who demonstrated that there is no consistent relationship between toxic behaviour and activity on a social media thread. Their results were based on a wide variety of topics and platforms spanning a timeframe from the 1990s to the 2020s. However, as the authors themselves note, the range of correlations between user activity and toxicity (from well below –.5 to well above .5) did not indicate a lack of effect of toxicity on engagement, but instead that a simple story of toxicity reducing activity is far from universal. Nevertheless, a number of limitations in their methods prevent a firm conclusion. First, it is possible that their results were confounded by a third, unmeasured variable. For example, it may be that perceived importance of the topic increased activity, including toxic activity, and the toxic behaviour (or expectations of toxic behaviour) reduced activity but generally not by enough to result in an obvious effect. Second, as their data were analysed by thread, not individual user, it is possible one group of individuals replaced others as the conversations became more toxic, which resulted in small correlations in aggregate. The latter could occur if toxic conversations attract “trolls” who enjoy such conversations, replacing others or unbalanced toxic activity could discourage activity unevenly for opposing opinions, resulting in increased activity on one side and a decline in the other side. Given that Avalle and colleagues [37] did not measure topic interest and did not distinguish toxic behaviour attacking one side from that of the other, neither of these possibilities can be ruled out based on their results. Alternatively, it is possible that the effect of toxic behaviour does not function at the thread or conversation level, instead reducing engagement over time, with little effect within conversations. Given evidence of individual-level effects of greater toxic behaviour on subsequent reductions in user activity in other research [33], the results of Avalle and colleagues [37] are interesting and informative, but do not rule out the potential of Spiral of Silence mechanisms to inform engagement on social media or the formation of echo chambers.”

Additionally, the inclusion of toxic interaction in the users' payoff function raises concerns, as it is not clear whether this might induce users to express opinions contrary to their beliefs. If this is the case, I believe the model could deviate significantly from reality, making its linkage to social psychology theories inappropriate.

Response:

For clarity, agents only share a view if their expected payoff is positive, and toxic behaviour can only reduce payoffs (provided cT > 0). In cases where payoffs are positive for both the pro- and anti-science positions, greater toxicity for one side than the other could tip the balance and this could mean agents would share information in favour of a position they are mildly on the other side of. However, the probability of this diminishes markedly as pS deviates from 0.5, depending on the values of various parameters. Based on a comment by Reviewer 2 (below), we have amended our model, and agents are now prevented from acting when the payoffs to both positions are within a margin determined by a new parameter, which further limits this possibility.

Secondly, the authors state that "to our knowledge, no model has incorporated insights from Spiral of Silence Theory and the concept of toxic interactions to explore opinion progression and echo chamber formation." This claim is only partially accurate. The concept of "hostile interaction” or “negative link” has already been explored in previous models (see, for example, Leskovec, Jure, Daniel Huttenlocher, and Jon Kleinberg. 'Predicting positive and negative links in online social networks.' Proceedings of the 19th international conference on World Wide Web. 2010).

Response:

Thank you for pointing out this issue. We have updated our text to better reflect why our model is unique, namely, that it allows both positive and negative (i.e., constructive and toxic) interactions between the same agents, meaning opposing agents can also persuade each other rather than only pushing each other apart.

The relevant paragraph (page 7, line 123) is:

“Although bounded confidence may be a reasonable description of some social media feed algorithms (i.e., the algorithms determining what content to show social media users) [51], they are unrealistic in most other scenarios. Bounded confidence models with dynamic networks either allow for very few interactions between agents with opposing views or they make all or most interactions between agents with opposing views negative. Evidence of toxic interactions between opposing groups [37,53] suggests that interactions between people with opposing views occur frequently and, although connections that can be broadly categorised as negative are not unreasonable [54,55], evidence of people being persuaded by civil interactions [14,56] suggests that not all interactions between those with opposing views are negative. Moreover, to date, negative interactions have been restricted to either dyadic interactions and relationships (i.e., an interaction between only two agents [54,55,57]) or a function of the receivers’ opinions relative to sharers’ opinions [18,37]. These two cases do not match the typical case of toxic online behaviour, whereby people choose to behave toxically, and that behaviour can be seen by anyone viewing the post (even if it is directed at a specific person), regardless of opinion. Therefore, the toxicity of a comment may be largely a function of the poster’s intent, not the receiver’s perception or the difference between poster and receiver’s opinion. Toxic behaviour, with a theoretical basis in Spiral of Silence Theory, may, therefore, be a more plausible mechanism than those previously employed as agents can interact with those with opposing views and disconnect from those who act toxically because they acted toxically, not because of interpretation due to opinion differences.”

The results of the model also require further clarification, as some outcomes seem implausible when compared to real-world scenarios. For instance, the authors claim that "61.0% of simulations resulted in all agents adopting the same view," which does not align with typical social media dynamics. This raises doubts about the model's relevance to the Pro/Anti-science debate, whether it accurately reflects any real life behavior and thus how it can be used to validate the hypothesized role of “Spiral of Silence” in shaping online debates.

Response:

The original goal of the simulations was to demonstrate that toxic behaviour working as a sanction could contribute to echo chamber formation via suppression of opposition opinions and persuasion of those who remained. This occurred in all simulations where toxic behaviour was present. However, it is implausible that everyone could come to believe or not believe something based on random factors and network dynamics, so we agree that it was difficult to position the model’s value. As a result, we have made changes to baseline parameters so that the model produces more realistic outcomes, and we have compared these to results from previously reported empirical social media data to confirm their realism and added this to the manuscript.

Moreover, the authors chose certain parameters as a baseline without providing sufficient justification. It remains unclear whether the model can reproduce real-life outcomes for specific parameter choices. The authors could strengthen their validation by comparing their results to empirical cases, as seen in Valensise, Carlo M., Matteo Cinelli, and Walter Quattrociocchi. 'The drivers of online polarization: Fitting models to data.' Information Sciences 642 (2023): 119152.

Response:

Thank you for the suggestion. We have followed the ideas of Valensise and colleagues (https://doi.org/10.1016/j.ins.2023.119152) and demonstrated that our model can match the general patterns of results reported in the earlier Cinelli and colleagues’ paper (https://doi.org/10.1073/pnas.2023301118). We have also provided a better description of the selection of baseline parameter values. The section where we describe the baseline parameter selections is reproduced below (from page 16, line 315) and the section where we compare our results to those of empirical data begins on page 26, line 570).

“A set of baseline parameter values and priors were chosen to enable an initial set of simulations to be performed. Values were selected based on bA = bAG = 1, and priors on toxic and constructive share rates of approximately 8% and 40%, respectively. Opinion priors were selected such that agents’ views would start distributed relatively evenly spread across the possible range (i.e., 0 to 1) but that they would be malleable within the model simulations and somewhat more malleable for those in the middle of the distribution. Other parameters were selected such that around 10-20% of agents would share in the first round (if they saw another agent’s post) prior to the addition of ejI and fij, with these rates approximately doubling after the addition of these random variables. Parameter values were fine-tuned to ensure the graph size did not grow too large and to avoid overly lop-sided results whereby everyone adopted a pro- or anti-science belief in the majority of simulations (at least in the baseline simulations).”

Another critical point requiring more attention is the exploration of the parameter space. As the authors note, they vary the toxicity parameter in increments of 0.5 within the interval [0,2]. However, in the range [1,2], the model exhibits minimal variation, with only one data point in the lower half of the range. I recommend increasing the granularity of this parameter sweep and offering a more in-depth explanation for the model's behavior.

Response:

In our update, we are no longer exploring the parameter space of cX > 1. Models with cX = 1 routinely produced the outcomes desired for high toxic activity, so higher values were not necessary. However, we have now included narrower increments at values close to 0 for cX and cT, where we expected more pronounced changes. We have also extensively updated our analysis of how and why changing parameters changes model outcomes.

Moreover, the statement that "this may have been because toxic propensity was sufficient for agents' views to effectively suppress opposing opinions for any value of cX ≥ 1" warrants further exploration. This analysis could provide insights into the model's transition behavior and the underlying cause of such changes.

Response:

Thank you for the comment. We have added further discussion of why there are limited changes in outcomes for higher values of cX (now ≥ 0.5) and cT.

The relevant sections (from page 23, line 497; page 24, line 511; and page 30, line 651) are reproduced below:

“At cT ≥ 0.5, reducing the value of cT made agents more willing to act in expectation of a particular level of toxic reprisal. In response, the opposing side became more likely to act toxically due to the presence of the share of agents supporting each side in the toxic behaviour equation (Equation (3)). As a result, changes to cT at higher levels simply adjusted th

---

## [Decision Letter · Decision Letter 1]

16 Apr 2025

Toxic behaviour facilitates echo chamber formation: An agent-based modelling simulation of science attitudes based on Spiral of Silence Theory

PONE-D-24-33055R1

Dear Dr. Bainbridge,

We’re pleased to inform you that your manuscript has been judged scientifically suitable for publication and will be formally accepted for publication once it meets all outstanding technical requirements.

Kind regards,

Fabiana Zollo, Ph.D.

Academic Editor

PLOS ONE

Additional Editor Comments (optional):

Reviewers' comments:

Reviewer's Responses to Questions

**Comments to the Author**

1. If the authors have adequately addressed your comments raised in a previous round of review and you feel that this manuscript is now acceptable for publication, you may indicate that here to bypass the “Comments to the Author” section, enter your conflict of interest statement in the “Confidential to Editor” section, and submit your "Accept" recommendation.

Reviewer #1: All comments have been addressed

Reviewer #2: All comments have been addressed

2. Is the manuscript technically sound, and do the data support the conclusions?

Reviewer #1: Yes

Reviewer #2: Yes

3. Has the statistical analysis been performed appropriately and rigorously? 

Reviewer #1: Yes

Reviewer #2: Yes

4. Have the authors made all data underlying the findings in their manuscript fully available?

Reviewer #1: Yes

Reviewer #2: Yes

5. Is the manuscript presented in an intelligible fashion and written in standard English?

Reviewer #1: Yes

Reviewer #2: Yes

6. Review Comments to the Author

Reviewer #1: I recognize the authors deeply revised their work and appreciate the effort they put in addressing all my comments.

Reviewer #2: The authors have made significant improvements to the paper, and I appreciate their thoughtful consideration of my comments.

In my view, the revised version meets the necessary standards and can now be accepted.

7. PLOS authors have the option to publish the peer review history of their article (what does this mean? ). If published, this will include your full peer review and any attached files.

**Do you want your identity to be public for this peer review?** For information about this choice, including consent withdrawal, please see our Privacy Policy .

Reviewer #1: No

Reviewer #2: No

---

## [Editor Report · Acceptance letter]

PONE-D-24-33055R1

PLOS ONE

Dear Dr. Bainbridge,

I'm pleased to inform you that your manuscript has been deemed suitable for publication in PLOS ONE. Congratulations! Your manuscript is now being handed over to our production team.

Kind regards,

on behalf of

Prof. Fabiana Zollo

Academic Editor

PLOS ONE